# The exchange dynamics of biomolecular condensates

Yaojun Zhang[1,2,3]*, Andrew GT Pyo[4], Ross Kliegman[2], Yoyo Jiang[2],
Clifford P Brangwynne[5,6], Howard A Stone[7], Ned S Wingreen[8,9]*

[1]Center for the Physics of Biological Function, Princeton University, Princeton, United
States; [2]Department of Physics and Astronomy, Johns Hopkins University, Baltimore,
United States; [3]Department of Biophysics, Johns Hopkins University, Baltimore,
United States; [4]Department of Physics, Princeton University, Princeton, United
States; [5]Department of Chemical and Biological Engineering, Princeton University,
Princeton, United States; [6]Howard Hughes Medical Institute, Chevy Chase, United
States; [7]Department of Mechanical and Aerospace Engineering, Princeton University,
Princeton, United States; [8]Department of Molecular Biology, Princeton University,
Princeton, United States; [9]Lewis-Sigler Institute for Integrative Genomics, Princeton,
United States

*For correspondence:
yaojunz@jhu.edu (YZ);
wingreen@princeton.edu (NSW)

Competing interest: See page
13

Reviewing Editor: Arvind
Murugan, University of Chicago,
United States

**Abstract** A hallmark of biomolecular condensates formed via liquid-liquid phase separation is
that they dynamically exchange material with their surroundings, and this process can be crucial
to condensate function. Intuitively, the rate of exchange can be limited by the flux from the dilute
phase or by the mixing speed in the dense phase. Surprisingly, a recent experiment suggests that
exchange can also be limited by the dynamics at the droplet interface, implying the existence of an
'interface resistance'. Here, we first derive an analytical expression for the timescale of condensate
material exchange, which clearly conveys the physical factors controlling exchange dynamics. We
then utilize sticker-spacer polymer models to show that interface resistance can arise when incident
molecules transiently touch the interface without entering the dense phase, i.e., the molecules
'bounce' from the interface. Our work provides insight into condensate exchange dynamics, with
implications for both natural and synthetic systems.

## eLife assessment

This **valuable** contribution studies factors that impact molecular exchange between dense and dilute
phases of biomolecular condensates through continuum models and coarse-grained simulations.
The authors provide **convincing** evidence that the bouncing of molecules off the interface can lead
to interfacial resistance and limit mixing. Results like these can inform how experimental results in
the field of biological condensates are interpreted.

## Introduction

The interior of cells is organized in both space and time by biomolecular condensates, which form
and dissolve as needed (*Shin and Brangwynne, 2017*; *Banani et al., 2017*). These condensates play
key roles in processes ranging from transcription to translation, metabolism, signaling, and more (*An
et al., 2008*; *Su et al., 2016*; *Sabari et al., 2018*; *Formicola et al., 2019*). The complex interactions
among their components endow condensates with distinct physical properties, including low surface
tension, viscoelasticity, aging, etc. These distinct properties are crucial to the ability of condensates to
carry out their biological functions. Here, we focus on one important physical property of condensates

**Figure 1.** FRAP experiments on droplets. (**A**) Schematic of a FRAP experiment in which an entire droplet is photobleached and the recovery of fluorescence is recorded. Material exchange between the condensate and the surrounding dilute phase can be limited by the flux of unbleached molecules coming from the dilute phase, the speed of internal mixing in the dense phase, or the flux passing through the interface. (**B**) The recovery time $\tau$ is defined as the time required for fluorescence to return 63% (i.e. $1 - 1/e$) of the way back to its original level. (**C**) Experimental data from **Taylor et al., 2019** in which a LAF-1 droplet of radius $R = 1\,\mu m$ recovers from photobleaching. (Left) Images before bleaching, immediately after bleaching of the entire droplet region, and at two subsequent times. (Right) Expected recovery times $\tau_{dil} = c_{den}R^2/(3c_{dil}D_{dil}) = 4.2 \pm 3.2\,s$ and $\tau_{den} = R^2/(\pi^2 D_{den}) = 60 \pm 18\,s$ if the slowest recovery process was either the flux from the dilute phase or diffusion within the droplet, respectively, with $D_{den} = 0.0017 \pm 0.0005\,\mu m^2/s$, $D_{dil} = 94 \pm 11\,\mu m^2/s$, and $c_{den}/c_{dil} = 1190 \pm 880$ taken from **Taylor et al., 2019**. While the timescale associated with interface resistance $\tau_{int}$ is unknown, the measured recovery time $\tau = 4570 \pm 470\,s$ is much longer than $\tau_{dil}$ and $\tau_{den}$, suggesting the recovery is limited by flux through the interface, with an interface conductance of $\kappa = R/3/(\tau - \tau_{dil} - \tau_{den}) = (7.4 \pm 0.8) \times 10^{-5}\,\mu m/s$.

– the rate of exchange of material between condensed and dilute phases. This rate of exchange can impact biochemical processes taking place in condensates by limiting the escape of completed products (e.g. ribosomes produced in nucleoli; **Yao et al., 2019**), or limiting the availability of components or regulatory molecules (e.g. snoRNAs and ribosomal proteins entering nucleoli, or mRNAs entering P bodies or stress granules). The rate of exchange can also control the dynamical response of condensates to a changing environment, and, as exchange between dense and dilute phase is central to coarsening via Ostwald ripening, it can regulate the number, size, and location of condensates within the cell.

The material exchange between a condensate and the surrounding dilute phase can be probed via FRAP experiments, a commonly used approach for measuring condensate fluidity and molecular diffusion coefficients. Exchange dynamics are thus readily measurable and have been reported for a variety of systems (**Li et al., 2012**; **Patel et al., 2015**; **Burke et al., 2015**; **Banani et al., 2016**; **Jain et al., 2016**; **Aumiller et al., 2016**). However, only a very limited number of studies (**Taylor et al., 2019**; **Hubatsch et al., 2021**; **Bo et al., 2021**; **Folkmann et al., 2021**; **Lee, 2021**) aimed to understand what controls the timescales of condensate component exchange. Briefly, **Taylor et al., 2019** combined FRAP experiments on condensates in vitro and in vivo with different theoretical models to examine the impact of model choice on the physical parameters derived from data fitting (**Taylor et al., 2019**). **Folkmann et al., 2021** and **Lee, 2021** proposed that the rate of molecular absorption to the condensate can be 'conversion-limited' instead of diffusion-limited and established a mathematical framework for the temporal evolution of droplet sizes in this limit (**Folkmann et al., 2021**; **Lee, 2021**). In all these cases, the modeling of interface resistance (**Taylor et al., 2019**) or conversion-limited material transfer (**Folkmann et al., 2021**; **Lee, 2021**) was conducted at the phenomenological level, without aiming to understand the underlying physical mechanism that gives rise to interface resistance. **Hubatsch et al., 2021** and **Bo et al., 2021** tackled the exchange dynamics problem by developing, respectively, a continuum theory of macroscopic phase separation (**Hubatsch et al., 2021**) and a stochastic Langevin equation of single-molecule trajectories (**Bo et al., 2021**). However, the mean-field approaches in **Hubatsch et al., 2021** and **Bo et al., 2021** neglect the potentially

complex dynamics of molecules at the condensate interface, which can slow down material exchange significantly as suggested by *Taylor et al., 2019* and *Folkmann et al., 2021*.

In the following, we first derive an analytical expression for the timescale of condensate material exchange, which conveys a clear physical picture of what controls this timescale. We then utilize a 'sticker-spacer' polymer model to investigate the mechanism of interface resistance. We find that a large interface resistance can occur when molecules bounce off the interface rather than being directly absorbed. We finally discuss the characteristic features of the FRAP recovery pattern of droplets when the exchange dynamics is limited by different factors.

## Results
### Mathematical formulation of exchange dynamics

The exchange of molecules between a condensate and the dilute phase can be investigated through FRAP-type experiments in which, e.g., fluorescence is locally bleached and recovery as a function of time recorded (*Figure 1A and B*). Theoretically, the time evolution of the concentration profile $c(r, t)$ of the molecules initially located in a spherical condensate of radius $R$ (bleached population) can be described by the following continuum diffusion equations (*Taylor et al., 2019*):

$$\frac{\partial}{\partial t} c(r, t) = D_{\text{den}} \left( \frac{\partial^2}{\partial r^2} + \frac{2}{r} \frac{\partial}{\partial r} \right) c(r, t), \quad r < R;$$
$$\frac{\partial}{\partial t} c(r, t) = D_{\text{dil}} \left( \frac{\partial^2}{\partial r^2} + \frac{2}{r} \frac{\partial}{\partial r} \right) c(r, t), \quad r > R,$$

(1)

with the initial condition:

$$c(r, 0) = \begin{cases} c_{\text{den}}, & r < R; \\ 0, & r > R, \end{cases}$$

(2)

and boundary conditions:

$$\frac{\partial}{\partial r} c(r, t) \bigg|_{r=0} = c(+\infty, t) = 0;$$
$$-D_{\text{den}} \frac{\partial}{\partial r} c(r, t) \bigg|_{r=R_-} = -D_{\text{dil}} \frac{\partial}{\partial r} c(r, t) \bigg|_{r=R_+} = \kappa \left[ c(R_-, t) - \frac{c_{\text{den}}}{c_{\text{dil}}} c(R_+, t) \right],$$

(3)

where $D_{\text{den}}$ and $D_{\text{dil}}$ are, respectively, the diffusion coefficients of molecules in the dense and dilute phases, and $c_{\text{den}}$ and $c_{\text{dil}}$ are, respectively, the equilibrium concentrations in the dense and dilute phases. The second boundary condition corresponds to flux balance at the interface of the condensate. Specifically, the flux exiting the dense phase (left) equals the flux entering the dilute phase (middle) and also equals the flux passing through the interface (right).

To understand the physical origin of the last term in the second boundary condition in *Equation 3*, we note that the net outward flux across the interface can be written as $k_- c(R_-, t) - k_+ c(R_+, t)$, where $k_{+/-}$ denotes the entering/exiting rate of molecules at the interface and $c(R_{+/-}, t)$ the concentration of bleached molecules immediately outside/inside of the boundary. At thermal equilibrium, this net flux goes to zero, i.e., $k_- c_{\text{den}} = k_+ c_{\text{dil}}$ so $k_+ = k_-(c_{\text{den}}/c_{\text{dil}})$. The net outward flux is therefore $k_- \left[ c(R_-, t) - (c_{\text{den}}/c_{\text{dil}}) c(R_+, t) \right]$. The parameter $\kappa \equiv k_-$ is a transfer coefficient that governs the magnitude of this net flux. When the ratio of the concentrations on the two sides of the interface deviates from the equilibrium ratio, a small $\kappa$ can kinetically limit the flux going through the interface. We therefore term $\kappa$ the interface conductance, the inverse of interface resistance.

For the model described by *Equations 1–3*, the fraction of molecules in the condensate which are unbleached at time $t$ is

$$f(t) = 1 - \frac{\int_0^R 4\pi r^2 c(r, t) dr}{\int_0^R 4\pi r^2 c_{\text{den}} dr}.$$

(4)

Clearly, how quickly $f(t)$ recovers from 0 to 1 quantifies the timescale of material exchange between the condensate and the surrounding dilute phase.

## Timescale of condensate component exchange

The authors of *Taylor et al., 2019* derived an exact solution for $f(t)$ in an integral form using Laplace transforms. However, it is not directly apparent from the integral expression what physics governs the timescale of fluorescence recovery. In addition, the lengthy integral form of the expression also presents an impediment to its practical experimental applications. To obtain a more intuitive and concise result, we note that diffusion of biomolecules in the dilute phase is typically much faster than diffusion in the dense phase, with measured $D_{dil}/D_{den}$ in the range of $10^2$–$10^5$ (*Freeman Rosenzweig et al., 2017*; *Taylor et al., 2019*). We therefore employed the exact solution to derive an approximate solution in the parameter regime $D_{dil} \gg D_{den}$:

$$f(t) = 1 - \exp\left(-\frac{t}{\tau}\right),$$ (5)

where the timescale of fluorescence recovery is given by

$$\tau = \frac{R^2}{\pi^2 D_{den}} + \frac{c_{den}R^2}{3c_{dil}D_{dil}} + \frac{R}{3\kappa}.$$ (6)

Please refer to Appendix 1 for a detailed derivation. We note that, in practice, $D_{dil} > 20D_{den}$ is sufficient for the validity of the approximation with the approximate $\tau$ in *Equation 6* within 10% of the exact value.

*Equation 6* conveys a clear physical picture of what controls the timescale of condensate material exchange. First, for large condensates and slow internal diffusion, exchange is limited by the rate of mixing within the condensate, so that $\tau \simeq R^2/(\pi^2 D_{den})$. Second, if instead diffusion in the dilute phase is sufficiently slow, or the concentration in the dilute phase is very low, then $\tau \simeq c_{den}R^2/(3c_{dil}D_{dil})$, which is the time required to replace all molecules in the condensate if molecules incident from the dilute phase are immediately absorbed (see Appendix 1). Finally, if the interface conductance $\kappa$ is very small, the interfacial flux can be rate limiting for exchange, yielding $\tau \simeq R/(3\kappa)$.

## Can interface resistance be much larger than predicted by mean-field theory?

What determines the magnitude of the interface conductance $\kappa$? From a theoretical perspective, transitions between dense and dilute phases have been modeled both from the continuum theory approach (*Hubatsch et al., 2021*) and by considering single-molecule trajectories (*Bo et al., 2021*). However, for any particular systems, the magnitude of the interface conductance depends on microscopic features of the biomolecules, such as internal states, which may not be captured by Flory-Huggins and Cahn-Hilliard-type mean-field theories. Indeed, if we start with the continuum approach in *Hubatsch et al., 2021*, where the concentration of bleached components $c(\mathbf{r}, t)$ is governed by

$$\frac{\partial c(\mathbf{r},t)}{\partial t} = \nabla \cdot \left\{ D\left[c_{eq}(\mathbf{r})\right] \left[\nabla c(\mathbf{r},t) - c(\mathbf{r},t)\frac{\nabla c_{eq}(\mathbf{r})}{c_{eq}(\mathbf{r})}\right] \right\}$$ (7)

with $c_{eq}(\mathbf{r})$ the equilibrium concentration profile and $D[c_{eq}(\mathbf{r})]$ the diffusion coefficient which depends on the local equilibrium concentration, one can obtain an expression for $\kappa$ (see Appendix 1):

$$\kappa^{-1} = \int \frac{c_{den}}{c_{eq}(r)D(r)}dr,$$ (8)

where the integral is over the interface region. We would then conclude $\kappa^{-1} < \delta/D_{den} + \delta c_{den}/(c_{dil}D_{dil})$, where $\delta$ is the width of the interface. As the interface is typically narrow, this inequality would imply that in practice the interfacial term in *Equation 6* would always be smaller than the sum of the other two terms, and thus could be neglected.

However, a recent FRAP experiment on LAF-1 protein droplets (*Taylor et al., 2019*) contradicts the above mean-field result. In the experiment, a micron-sized LAF-1 droplet ($R = 1 \mu m$) was bleached and fluorescence recovery measured as a function of time (*Figure 1C*). It was observed that recovery of that

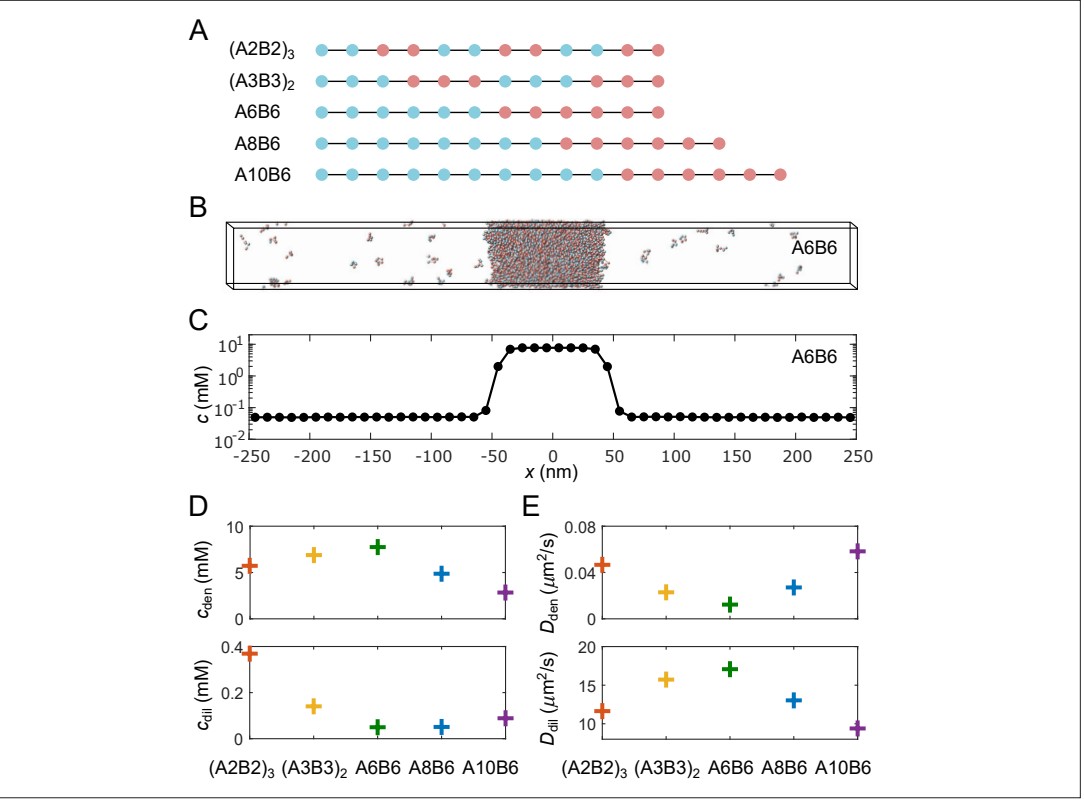

**Figure 2.** Coarse-grained molecular-dynamics simulations of multivalent phase-separating polymers. (**A**) Each polymer is composed of monomers ('stickers') of type A (blue) or B (red), and modeled as a linear chain of spherical particles each with a diameter of 2 nm, connected by stretchable bonds with an equilibrium length of 3.9 nm. Stickers of different types have an attractive interaction, while stickers of the same type interact repulsively, ensuring one-to-one binding between the A and B stickers. (**B**) Snapshot of a simulation of 1000 A6B6 polymers in a 500 nm × 50 nm × 50 nm box with periodic boundary conditions. The system undergoes phase separation into a dense phase (middle region) and a dilute phase (two sides), driven by the one-to-one A-B bonds. (**C**) Polymer concentration profile for the simulation in (**B**) with the center of the dense phase aligned at $x = 0$ and averaged over time and over 10 simulation repeats. (**D**) Average total polymer concentrations in the dense (top) and dilute (bottom) phases from simulations of the five types of polymers shown in (**A**). (**E**) Polymer diffusion coefficients in the dense (top) and dilute (bottom) phases. All simulations were performed and snapshots were obtained using LAMMPS *Plimpton, 1995*. Please refer to Appendix 2 for simulation details.

droplet occurs on a timescale of ~ 1.3hr. Given the measured parameters of the system, one can estimate the recovery time in the mean-field approach to be $\tau = R^2/(\pi^2 D_{\text{den}}) + c_{\text{den}}R^2/(3c_{\text{dil}}D_{\text{dil}}) = 64 \pm 18\,\text{s}$, much shorter than the measured recovery time. A large interface resistance was proposed as a possible explanation for this discrepancy (*Taylor et al., 2019*). Motivated by this surprising experimental result, we sought to investigate if it is possible for the interface resistance to be much larger than predicted by mean-field theory, and if so, what could be the underlying mechanisms and how does the interface resistance depend on the microscopic features of phase-separating molecules?

## Coarse-grained simulation of 'sticker-spacer' polymer phase separation

As noted above, if all molecules incident from the dilute phase are immediately absorbed into the dense phase, the interfacial flux can't be rate limiting. The existence of a large interface resistance then necessarily implies a strongly reduced flux of molecules successfully crossing the interface. This can occur either because the molecules incident from the dilute phase fail to incorporate into the interface, or they transiently incorporate but fail to enter the dense phase. In both cases, the molecules effectively 'bounce' from the interface leading to a large interface resistance. Mechanistically, bouncing can occur for a variety of reasons, which we discuss in the Discussion section below. Here, we employ a 'sticker-spacer' polymer model (*Choi et al., 2020*; *Semenov and Rubinstein, 1998*) to

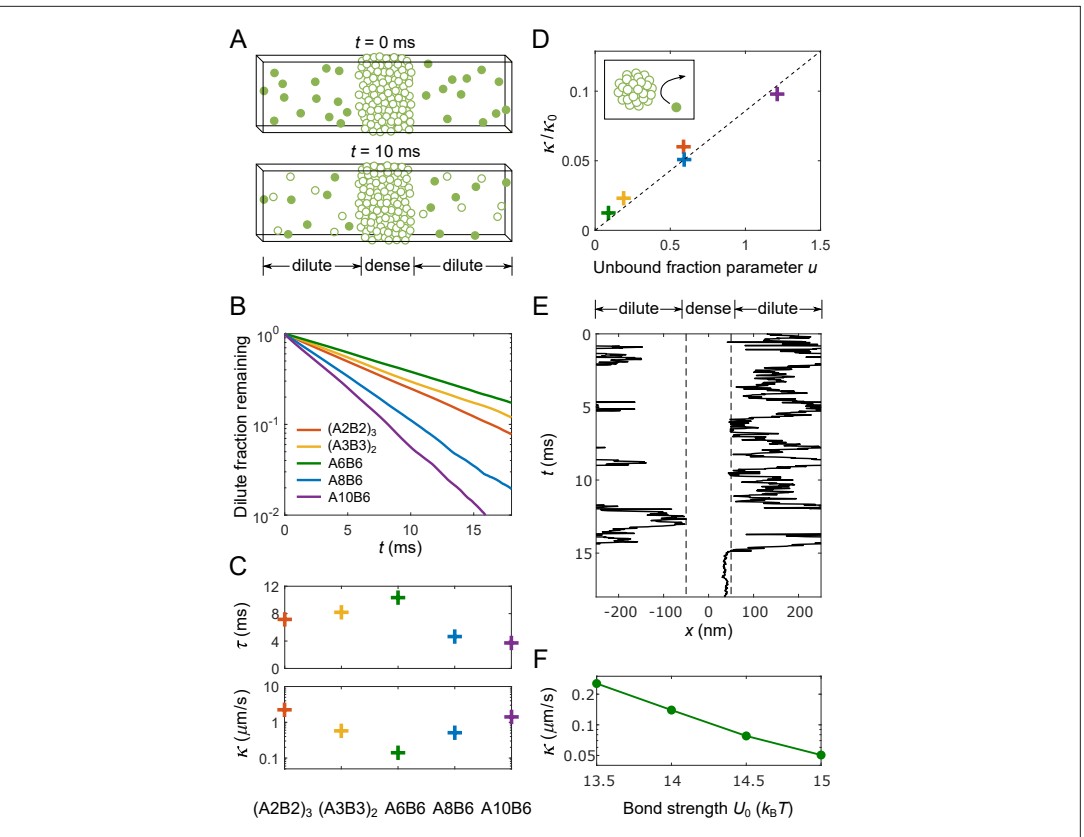

**Figure 3.** Determination of interface conductance from simulations. (**A**) Illustration of simulation protocol: At $t = 0$ only polymers in the dilute phase are 'labeled' (solid balls), any polymer that enters the dense phase (forms an A-B bond lasting >10 times the average bond lifetime of an isolated A-B pair) becomes permanently 'unlabeled' (hollow balls). (**B**) Fraction of labeled polymers in the dilute phase as a function of time for simulations of the five types of polymers shown in *Figure 2A*. (**C**) Decay time of labeled polymers from exponential fits to curves in (**B**) (top), and corresponding calculated values of interface conductance $\kappa$ (bottom). (**D**) For all simulated polymers, interface conductance scaled by $\kappa_0$ (*Equation 13*) is approximately a linear function of a parameter $u$ which reflects the fraction of unbound stickers in the dense and dilute phases. Inset illustration: Polymers in the dilute phase with few or no unbound stickers may 'bounce' off the dense phase, which contributes to the interface resistance. (**E**) Example of simulated trajectory in which a dilute-phase A6B6 polymer 'bounces' multiple times before finally joining the dense phase. (**F**) Interface conductance $\kappa$ of A6B6 system as a function of binding strength $U_0$ between A and B stickers.

explore one possible mechanism in which molecules can assume non-sticking conformations by saturating all their possible binding sites. These molecules incident from the dilute phase typically fail to form bonds with the dense phase, thus 'bouncing' off of the condensate.

The 'sticker-spacer' model provides a conceptual framework for understanding biomolecular phase separation, wherein the 'stickers' represent residues or larger domains that are capable of forming saturable bonds, while the 'spacers' connect the stickers to form polymers. Specifically, we simulated polymers consisting of type A and type B stickers connected by implicit spacers in the form of stretchable bonds (*Kremer and Grest, 1990*; *Figure 2A*):

$$U_{\mathrm{b}}(r) = -\frac{1}{2}KR_0^2 \ln\left[1 - \left(\frac{r}{R_0}\right)^2\right], \quad r < R_0, \tag{9}$$

where $r$ is the distance between two stickers. One-to-one heterotypic bonds between A and B are implemented via an attractive potential:

$$U_{\mathrm{a}}(r) = -\frac{1}{2}U_0\left(1 + \cos\frac{\pi r}{r_0}\right), \quad r < r_0, \tag{10}$$

while stickers of the same type interact through a repulsive potential to prevent many-to-one binding:

$$U_\mathrm{r}(r) = 4\epsilon\left[\left(\frac{\sigma}{r}\right)^{12} - \left(\frac{\sigma}{r}\right)^{6}\right] + \epsilon, \quad r \le r_\mathrm{c}. \tag{11}$$

We take $K = 0.15\,k_\mathrm{B}T/\mathrm{nm}^2$, $R_0 = 10\,\mathrm{nm}$, $U_0 = 14\,k_\mathrm{B}T$, $r_0 = 1\,\mathrm{nm}$, $\epsilon = 1\,k_\mathrm{B}T$, $\sigma = 2\,\mathrm{nm}$, and $r_\mathrm{c} = 1.12\sigma$ in all simulations, except in the simulations of **Figure 3F** where we vary $U_0$ systematically from 13.5 to $15\,k_\mathrm{B}T$. For all simulation results we reported below, the standard error of the mean is typically smaller than the symbol size and therefore not shown.

For each of the five sequences shown in **Figure 2A**, we simulated 1000 polymers in a $500\,\mathrm{nm} \times 50\,\mathrm{nm} \times 50\,\mathrm{nm}$ box with periodic boundary conditions using Langevin dynamics (see Appendix 2 for details). Simulations were performed using LAMMPS molecular-dynamics simulator (**Plimpton, 1995**). **Figure 2B** shows a snapshot of coexisting dense and dilute phases after equilibration of the A6B6 polymers (6A stickers followed by 6B stickers), while **Figure 2C** shows the time-averaged profile of the total polymer concentration. The five different polymer sequences we simulated were chosen to yield a range of dilute- and dense-phase sticker concentrations (**Figure 2D**) as well as a range of dilute- and dense-phase diffusion coefficients (**Figure 2E**). As found previously (**Weiner et al., 2021**), polymers like A6B6 with long blocks of stickers of the same type have low dilute-phase concentrations. This follows because it is entropically unfavorable for these polymers to form multiple self-bonds, which favors the dense phase where these polymers can readily form multiple trans-bonds. These long-block polymers also have low dense-phase diffusion coefficients because of their large number of trans-bonds, which need to be repeatedly broken for the polymers to diffuse.

## Interface conductance $\kappa$ from simulations

Having determined the concentrations and diffusion coefficients in the dense and dilute phases, we are now in a position to extract the values of interface conductance from simulations. **Figure 3** depicts a simple protocol that allows us to infer $\kappa$ by applying the 1D, slab-geometry version of **Equations 1–6** to simulation results (see Appendices 1 and 2 for details): (i) All polymers in the dilute phase are initially considered 'labeled', (ii) any labeled polymer that forms a lasting A-B bond with a polymer in the dense phase becomes permanently unlabeled (**Figure 3A**), (iii) the remaining fraction of labeled dilute phase polymers is fit to an exponential decay (**Figure 3B**), and (iv) the resulting decay time constant $\tau$ is used together with the known dense and dilute phase parameters to infer $\kappa$ from:

$$\kappa = \frac{c_\mathrm{dil}}{c_\mathrm{den}} \sqrt{\frac{D_\mathrm{dil}}{\tau}} \tan \frac{d}{\sqrt{\tau D_\mathrm{dil}}}, \tag{12}$$

where $d$ is the half-width of the dilute phase. As shown in **Figure 3C**, the resulting values of $\kappa$ span more than an order of magnitude for our selected polymer sequences, despite the fact that all five polymers can in principle form the same number (6) of self-bonds.

We note that one can alternatively obtain $\kappa$ by directly measuring the flux of molecules that enter the dense phase. Mathematically, this flux equals $k_+ c_\mathrm{dil} = \kappa c_\mathrm{den}$. We show in Appendix 2 that the values of $\kappa$ found via this method are consistent with results reported in **Figure 3C**.

## 'Bouncing' of molecules can lead to large interface resistance

What gives rise to the very different values of $\kappa$? To address this question, we first consider the predicted interface conductance $\kappa_0$ if polymers incident from the dilute phase simply move through the interface region with a local diffusion coefficient that crosses over from $D_\mathrm{dil}$ to $D_\mathrm{den}$. Then according to **Equation 8** (see Appendix 1)

$$\kappa_0 = \frac{c_\mathrm{dil} D_\mathrm{dil}}{\delta c_\mathrm{den}}. \tag{13}$$

However, as shown in **Figure 3D**, the actual values of $\kappa$ in our simulations can be a factor of ~50 smaller than $\kappa_0$. This reduction can be traced to a 'bouncing' effect. As shown schematically in the inset to **Figure 3D** and for an exemplary simulated trajectory in **Figure 3E** (more trajectories can be found in Appendix 2), molecules incident from the dilute phase may fail to form bonds with the dense phase, effectively 'bouncing' off of the condensate. The differing extent of this bouncing effect for the

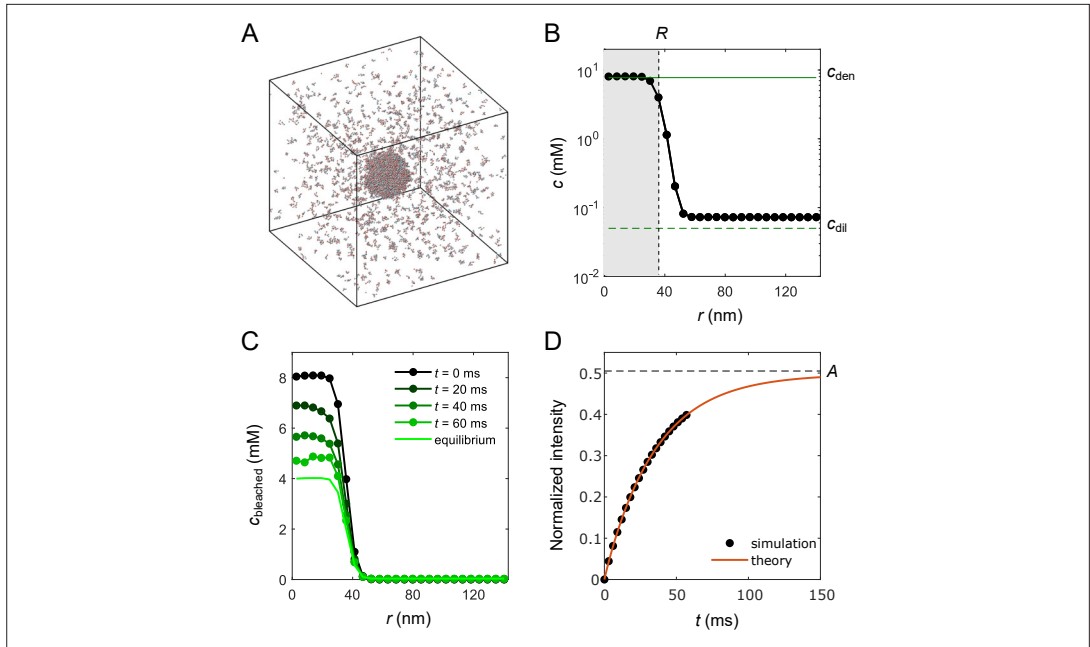

**Figure 4.** Molecular-dynamics simulations of in silico 'FRAP experiment' on a small droplet of A6B6 polymers. (**A**) Snapshot of 2000 A6B6 polymers in a 286 nm × 286 nm × 286 nm box with periodic boundary conditions. The system phase separates into a dense droplet (middle) and a surrounding dilute phase. (**B**) Polymer concentration profile for the simulation in (**A**) with the center of the droplet aligned to the origin and averaged over time and over 10 simulation repeats. The dilute- and dense-phase concentrations obtained from simulations in the slab geometry are denoted by green lines. The effective radius $R$ of the droplet is determined through $4\pi R^3 c_{\text{den}}^{\text{d}}/3 + (V_{\text{box}} - 4\pi R^3/3)c_{\text{dil}}^{\text{d}} = N_{\text{total}}$, where $V_{\text{box}}$ is the volume of the simulation box and $N_{\text{total}}$ is the total number of polymers. (**C**) Average concentration profile of bleached population measured at different times. (**D**) Comparison of the FRAP recovery curves obtained by tracking the simulated fraction of unbleached molecules inside the droplet as a function of time (black dots) and by numerical integration (red curve) of **Equation 1** in a sphere of the same volume as $V_{\text{box}}$ using the measured parameters of the droplet system: $c_{\text{den}}^{\text{d}} = 8.1\,\text{mM}$, $c_{\text{dil}}^{\text{d}} = 0.073\,\text{mM}$, $D_{\text{den}} = 0.013\,\mu\text{m}^2/\text{s}$, $D_{\text{dil}} = 17\,\mu\text{m}^2/\text{s}$, $R = 0.037\,\mu\text{m}$, and $\kappa^{\text{d}} = 0.20\,\mu\text{m/s}$. For details of simulation and theory, refer to Appendix 2.

five sequences we studied reflects differences in their numbers of free stickers in both their dilute- and dense-phase conformations. The fewer such available stickers, the fewer ways for a polymer incident from the dilute phase to bond with polymers at the surface of the dense phase, and thus the more likely the incident polymer is to bounce. More generally, we find that the interface conductance of the sticker-spacer polymers is controlled by the encounter rate of a pair of unbound stickers and the availability of these stickers, which in turn depends on the sticker-sticker binding strength, the dilute- and dense-phase polymer concentrations, and the width of the interface:

$$\kappa = \frac{4\pi r_0 \delta D_{\text{dil}} n^2 c_{\text{dil}}}{2 + s + s^{-1}} \left(f_{\text{dilA}}f_{\text{denB}} + f_{\text{dilB}}f_{\text{denA}}\right), \tag{14}$$

where $n$ is the number of monomers in a polymer, $s$ is the global stoichiometry (i.e. $c_{\text{A}}/c_{\text{B}}$), $f_{\text{dilA/dilB}}$ and $f_{\text{denA/denB}}$ are the fractions of unbound A/B monomers in the dilute and dense phases, respectively. In support of this picture, we find that all our simulation results for $\kappa/\kappa_0$ collapse as a linear function of a lumped parameter $u$ (**Figure 3D**):

$$u = \frac{4\pi r_0 \delta^2 n^2 c_{\text{den}}}{2 + s + s^{-1}} (f_{\text{dilA}}f_{\text{denB}} + f_{\text{dilB}}f_{\text{denA}}), \tag{15}$$

which expresses the availability of free stickers, where all parameters in $u$ are determined directly from simulations. See Appendix 1 for derivations of **Equations 14 and 15**.

Comparing sequences with unequal sticker stoichiometry A8B6 and A10B6 to their most closely related equal-stoichiometry sequence A6B6, we find that the extra A stickers substantially increase

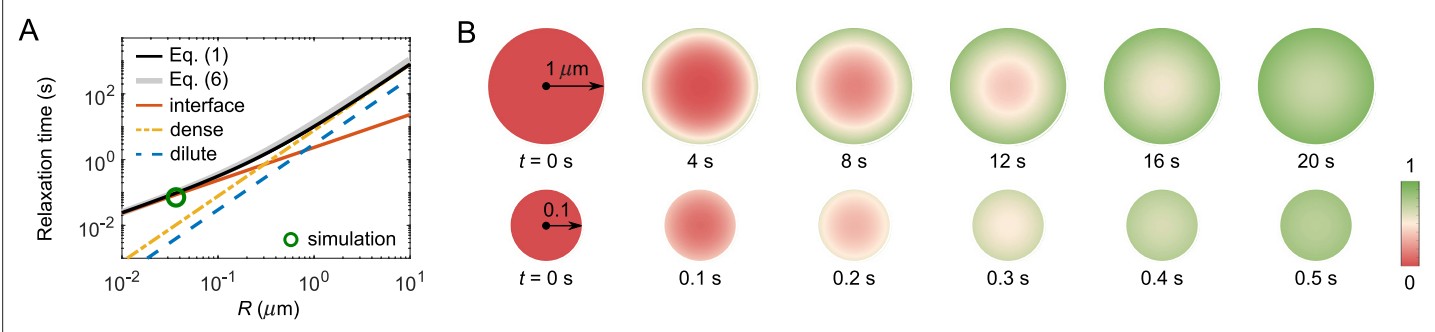

**Figure 5.** FRAP recovery patterns for large versus small droplets can be notably different for condensates with a sufficiently large interface resistance.
(**A**) Expected relaxation time as a function of droplet radius for in silico 'FRAP experiments' on the A6B6 system. The interface resistance dominates recovery times for smaller droplets, whereas dense-phase diffusion dominates recovery times for larger droplets. Green circle: FRAP recovery time obtained from direct simulation of an A6B6 droplet of radius 37 nm. Black curve: the recovery time as a function of droplet radius from a single exponential fit of the exact solution of the recovery curve from *Equation 1–4*. Gray curve: the recovery time predicted by *Equation 6*. Yellow, blue, and red curves: the recovery time when dense-phase, dilute-phase, and interface flux limit the exchange dynamics, i.e., the first, second, and last term in *Equation 6*, respectively. Parameters matched to the simulated A6B6 system in the slab geometry: $c_{\mathrm{den}} = 7.7\,\mathrm{mM}$, $c_{\mathrm{dil}} = 0.05\,\mathrm{mM}$, $D_{\mathrm{den}} = 0.013\,\mu\mathrm{m}^2/\mathrm{s}$, $D_{\mathrm{dil}} = 17\,\mu\mathrm{m}^2/\mathrm{s}$, and $\kappa = 0.14\,\mu\mathrm{m/s}$. (**B**) Time courses of fluorescence profiles for A6B6 droplets of radius $1\,\mu\mathrm{m}$ (top) and $0.1\,\mu\mathrm{m}$ (bottom); red is fully bleached, green is fully recovered. These concentration profiles are the numerical solutions of *Equations 1–3* using parameters provided in (**A**).

the interface conductance $\kappa$. Intuitively, the excess As in both dense and dilute phases of A8B6 and A10B6 provide a pool of available stickers for any unbound B to bind to. By contrast, at equal stoichiometry, both free As and free Bs are rare which maximizes the bouncing effect. This reduction in potential binding partners at equal stoichiometry has also been observed experimentally (*Brassinne et al., 2017*), and theoretically *Ronceray et al., 2022*, to cause an anomalous slowing of diffusion within condensates at equal stoichiometry in the regime of strong binding.

Finally, we expect the interface resistance to increase approximately exponentially with the increase of binding strength $U_0$ between A and B stickers, as the tighter the binding, the fewer available stickers, and hence the more bouncing of molecules at the interface. We demonstrate in *Figure 3F* that the interface conductance $\kappa$ of the A6B6 system indeed drops by a factor of 5 as the value of $U_0$ increases from 13.5 to $15\,k_{\mathrm{B}}T$.

## Direct simulation of droplet FRAP

Above we simulated phase separation of sticker-spacer polymers in a slab geometry, and discussed how the extracted interface conductance $\kappa$ depends on sequence pattern, sticker stoichiometry, and binding strength between stickers. In principle, with the parameters measured from such simulations, *Equation 6* and *Equations 1–4* can be used to predict FRAP recovery times for simulated 3D droplets. To check the consistency between theory and simulation, we simulated a small droplet of the A6B6 polymers and measured its FRAP recovery time. Briefly, we simulated 2000 A6B6 polymers in a cubic box of side length $286\,\mathrm{nm}$ with periodic boundaries using Langevin dynamics. All interaction potentials and parameters are the same as the simulations in *Figure 2*. *Figure 4A* shows a snapshot of the droplet coexisting with the surrounding dilute phase after equilibration. *Figure 4B* shows the mean polymer concentration profile, which is consistent with the dense- and dilute-phase concentrations of the A6B6 system reported in *Figure 2D*. We note that both concentrations are slightly higher than their counterparts in the slab geometry due to a surface tension effect (*Thomson, 1872*). At time $t = 0$, we labeled all the molecules inside the droplet as 'bleached' and tracked the time evolution of the concentration profile of the bleached population (*Figure 4C*) and obtained the FRAP recovery curve (*Figure 4D*, black circle). We note that the recovery curve plateaus at $A \approx 0.5$ instead of 1 due to limited number of polymers in the dilute phase. Using parameters of the droplet system, we numerically integrated *Equation 1* with modified initial and boundary conditions to account for the system's finite size. The resulting numerical curve agrees almost perfectly with the simulation result (*Figure 4D*), which validates both our theoretical and simulation approaches.

Fitting the recovery curve by

$$f(t) = A \left[ 1 - \exp\left( -\frac{t}{A\tau} \right) \right], \tag{16}$$

which takes into account the effect of the finite-size dilute phase (see Appendix 2 for a derivation), yielded a recovery time $\tau = 0.071\,\text{s}$. We compare the measured FRAP recovery time for the small droplet $R = 37\,\text{nm}$ (green circle) to theoretical predictions from *Equation 6* (gray) and *Equations 1–4* (black) in *Figure 5A*. The FRAP recovery of the simulated droplet is clearly limited by the interface resistance. We note that the small deviation between theory and simulation in *Figure 5A* is due to the utilization of parameters from the slab geometry for the theory prediction, including a $\kappa = 0.14\,\mu\text{m/s}$ lower than the measured $\kappa^{\text{d}} = 0.20\,\mu\text{m/s}$ of the droplet system, which in turn reflects the difference in the dilute-phase concentrations of the two systems, as $\kappa \sim c_{\text{dil}}$ from *Equation 14*.

## Signatures of interface resistance

Under what circumstances is interface resistance experimentally measurable? If there were no bouncing effect, i.e., if all molecules incident from the dilute phase that touch the interface get immediately absorbed into the condensate, then interface resistance would never dominate the recovery time in FRAP-type experiments, making it very difficult to measure $\kappa$. However, as shown in *Figure 3*, the bouncing effect can reduce $\kappa$ substantially. For such systems, the interface conductance can be inferred quantitatively from *Equation 6* or by fitting FRAP recovery curves as in *Taylor et al., 2019*, using the experimentally measured dense- and dilute-phase concentrations and diffusion coefficients.

Even without knowing all parameters, one may still be able to infer the presence of a large interface resistance by observing the pattern of fluorescence recovery in droplets of different sizes. According to *Equation 6* the recovery time associated with interface resistance increases linearly with radius $R$ while the other terms increase as $R^2$ (*Figure 5A*). Therefore, one expects a cross-over for the recovery from being interface-resistance dominated (small $R$) to being either dilute-phase-diffusion or dense-phase-mixing dominated (large $R$). In the latter case, the fluorescence profile during recovery will be notably different in large versus small droplets as shown in *Figure 5B* – for large droplets progressive diffusion of fluorescence into the droplet will be apparent, whereas small droplets will recover uniformly as internal mixing will be fast compared to exchange with the surroundings. Thus observation of such a cross-over of the recovery pattern as a function of droplet size provides evidence for the presence of a large interface resistance, which can be followed up by more quantitative studies. For example, the uniform recovery of the LAF-1 droplet in *Figure 1C* and the simulated droplet in *Figure 4C* are indicative of a large interface resistance, as the diffusion in the dilute phase is too fast to be rate limiting. We also predict the cross-over for LAF-1 droplets to be around $R = 71\,\mu\text{m}$, which in principle can be tested experimentally.

## Discussion

The dynamic exchange of condensate components with the surroundings is a key feature of membraneless organelles, and can significantly impact condensate biological function. In this work, we combined analytical theory and coarse-grained simulations to uncover physical mechanisms that can control this exchange dynamics. Specifically, we first derived an analytical expression for the exchange rate, which conveys the clear physical picture that this rate can be limited by the flux of molecules from the dilute phase, by the speed of mixing inside the dense phase, or by the dynamics of molecules at the droplet interface. Motivated by recent FRAP measurements (*Taylor et al., 2019*) that the exchange rate of LAF-1 droplets can be limited by interface resistance, which contradicts predictions of conventional mean-field theory, we investigated possible physical mechanisms underlying interface resistance using a 'sticker-spacer' model. Specifically, we demonstrated via simulations a notable example in which incident molecules have formed all possible internal bonds, and thus bounce from the interface, giving rise to a large interface resistance. Finally, we discussed the signatures in FRAP recovery patterns when the exchange dynamics is limited by different factors.

What are potential mechanisms that could lead to the bouncing of molecules from the interface and hence to a substantial interface resistance? The essential requirement is that molecules in the dilute phase and molecules at the interface should not present 'sticky' surfaces to each other. Since these same molecules must be capable of sticking to each other in order to phase separate, a natural scenario is that these molecules assume non-sticky conformations due to the shielding of interacting

regions, e.g., burial of hydrophobic residues in the core of a protein, or, in the scenario explored in the simulations, the saturation of sticker-like bonds. Examples of systems with strong enough bonds to allow bond saturation include SIM-SUMO (*Banani et al., 2016*) and nucleic acids with strong intramolecular base-pairing. Interestingly, a recent coarse-grained simulation of RNA droplets of $(CAG)_{47}$ (*Nguyen et al., 2022*) illustrated that a $(CAG)_{47}$ molecule in a closed hairpin conformation fails to integrate into a droplet but rather bounces off the droplet interface. Another possible scenario is that charged molecules could arrange themselves to form a charged layer at the interface, resulting in a high energetic barrier from electrostatic repulsion for a dilute-phase component to reach and cross the interface (*Ray et al., 2023*; *Dai et al., 2023*; *Majee et al., 2024*). In the case of LAF-1, we note that the values of interface conductance $\kappa$ obtained in our simulations are a factor of $10^3$ to $10^4$ higher than the experimentally measured $\kappa$ for the LAF-1 droplet. While we do not aim to specifically simulate the LAF-1 system in this work and the value of $\kappa$ in simulations can in principle be tuned by adjusting the bond strength $U_0$, the large disparity between simulation and experiment renders the mechanism responsible for the inferred large interface resistance in LAF-1 droplets unclear. We hope that our study will motivate further experimental investigations into the anomalous exchange dynamics of LAF-1 droplets and potentially other condensates, and the mechanisms underlying interface resistance.

In this work, we focused on the exchange dynamics of in vitro single-component condensates. How is the picture modified for condensates inside cells? It has been shown that Ddx4-YFP droplets in the cell nucleus exhibit negligible interface resistance (*Taylor et al., 2019*), which raises the question whether interface resistance is relevant to natural condensates in vivo. Future quantitative FRAP and single-molecule tracking experiments on different types of droplets in the cell will address this question. One complication is that condensates in cells are almost always multi-component, which can increase the complexity of the exchange dynamics. Interestingly, formation of multiple layers or the presence of excess molecules of one species coating the droplet is likely to increase interface resistance. A notable example is the Pickering effect, in which adsorbed particles partially cover the interface, thereby reducing the accessible area and the overall condensate surface tension, slowing down the exchange dynamics (*Folkmann et al., 2021*). The development of theory and modeling for the exchange dynamics of multi-component condensates is currently underway.

Biologically, the interface exchange dynamics also influences the coarsening of condensates. The same interface resistance that governs exchange between phases at equilibrium will control the flux of material from the dilute phase to the dense phase during coarsening, so that bouncing will slow down the coarsening process. Indeed, a recent theoretical study (*Ranganathan and Shakhnovich, 2020*) of coarsening via mergers of small polymer clusters found anomalously slow coarsening dynamics due to exhaustion of binding sites, paralleling the single-polymer bouncing effect explored here. Other mechanisms that may slow coarsening include the formation of metastable microemulsions (*Welsh et al., 2022*; *Kelley et al., 2021*) and the Pickering effect (*Folkmann et al., 2021*) mentioned above. In the latter study, additional slow coarsening of PGL-3 condensates was attributed to a conversion-limited (i.e. interface resistance) rather than a diffusion-limited flux of particles from the dilute phase into the dense phase. Interestingly, a conversion-limited flux has been shown to lead to qualitatively distinct scaling of condensate size with time (*Lee, 2021*). As many condensates dissolve and reform every cell cycle (or as needed), we anticipate that interfacial exchange will constitute an additional means of regulating condensate dynamics.

## Methods

We perform coarse-grained molecular-dynamics simulations using LAMMPS (*Plimpton, 1995*) to simulate phase separation of 'sticker and spacer' polymers. Individual polymers are modeled as linear chains of spherical stickers of types A and B connected by implicit spacers (*Figure 2A*) with the interaction potentials in *Equations 9–11*, which ensure one-to-one binding between A and B stickers. For each of the five selected polymer sequences, we perform 10 simulation replicates with different random seeds in a slab geometry. Consistency of results is checked across replicates and across the first and second halves of the recorded data. The agreement indicates that the system has reached equilibrium. For details see Appendix 2, Simulation procedures and data recording.

To measure the dilute- and dense-phase concentrations, we first group polymers into connected clusters in each recording. Two stickers are considered connected if they are part of the same polymer,

or if they are within the attraction distance $r_0 = 1\,\mathrm{nm}$. Connected stickers are then grouped into clusters. To find the concentrations of each phase, we identify the center of mass of the largest cluster in each recording, and recenter the simulation box to this center of mass. The resulting polymer concentration profile has high values in the middle corresponding to the dense-phase concentration, and low values on the two sides corresponding to the dilute-phase concentration (*Figure 2C*). For details, see Appendix 2, Determining the dilute- and dense-phase concentrations.

To measure the dilute- and dense-phase diffusion coefficients, we perform simulations with a pure dilute phase or dense phase, i.e., with polymers at the measured dilute- and dense-phase concentrations. To find the diffusion coefficients, we compute the time-averaged mean squared displacement (MSD) for each polymer as a function of the lag time $t_{\mathrm{lag}}$, and average over all polymers in a simulation box and over five replicates. The time- and ensemble-averaged MSD is then linearly fit to $\mathrm{MSD} = 6Dt_{\mathrm{lag}}$ to extract the diffusion coefficient. For details, see Appendix 2, Determining the dilute- and dense-phase diffusion coefficients.

To measure the interface conductance $\kappa$, we follow the simple protocol depicted in *Figure 3A*. Specifically, in this protocol we first define a 'survival' variable $S$ for each polymer in the dilute phase as a function of time: $S = 1$ if the polymer has remained in the dilute phase, and $S = 0$ if the polymer has ever entered the dense-phase cluster. The obtained $S(t)$ is the average survival probability of polymers in the dilute phase that have never entered the dense phase. We fit $S(t)$ to a decaying exponential to extract the decay time $\tau$. The interface conductance $\kappa$ is then calculated using *Equation 12* with the measured decay time and dilute- and dense-phase parameters. For details, see Appendix 2, Determining the interface conductance.

Simulations of the A6B6 spherical droplet system largely follow their counterparts in the slab geometry. To obtain the concentration profile in *Figure 4B*, we identify the center of mass of the droplet and recenter the simulation box to this center of mass in each recording. We then compute the time- and ensemble-averaged polymer concentration histogram along the radial direction. The dilute- and dense-phase concentrations ($c_{\mathrm{dil}}^{\mathrm{d}}$ and $c_{\mathrm{den}}^{\mathrm{d}}$) of the droplet system are calculated by averaging the concentration profile over the relevant regions. To obtain the concentration profile of the bleached population at time $t$ after photobleaching in *Figure 4C*, we label all polymers in the droplet at time $t_0$ as bleached and track the concentration profile of these polymers at a later time $t_0 + t$. Results are averaged over all possible choices of $t_0$. To obtain the theory curve in *Figure 4D*, we numerically integrate *Equation 1* using a finite-difference method. Interface conductance of the droplet system is determined using the flux method. For details, see Appendix 2, Details of simulation and theory of FRAP recovery of an A6B6 droplet.

The codes for generating simulated data following the above-mentioned methods are uploaded as *Source code 1*. *Source code 1* contains MATLAB codes used to generate input files for the LAMMPS Molecular Dynamics Simulator, as well as the generated LAMMPS input files. All data in the manuscript can be reproduced using these files. LAMMPS input files are contained in the folders: FullSystem, DensePhase, and DilutePhase. Codes in FullSystem/In are for simulations in slab geometry at a fixed interaction strength $U_0 = 14\,k_{\mathrm{B}}T$, which generate the data shown in *Figures 2 and 3*. Codes in FullSystem/In_A are for simulations in slab geometry at varying interaction strengths, which generate the data shown in *Figure 2F*. Codes in FullSystem/In_Droplet are for simulations of a 3D droplet, which generate the data shown in *Figure 4*. Codes in DensePhase and DilutePhase are used to measure the diffusion coefficients of molecules in dense and dilute phases, respectively, which generate the data shown in *Figure 2E*.

## Acknowledgements

This work was supported in part by the National Science Foundation, through the Center for the Physics of Biological Function (PHY-1734030), NIH Grants R01 GM140032, the Howard Hughes Medical Institute, and the Air Force Office of Scientific Research (FA9550-20-1-0241 to CPB). YZ and RK were partially supported by a startup fund at Johns Hopkins University. CPB and HAS were partially supported by Princeton University's Materials Research Science and Engineering Center DMR-1420541. We also thank the Princeton Biomolecular Condensate Program for funding support. The authors acknowledge that the work reported in this paper was performed using the Princeton Research Computing resources at Princeton University and the Advanced Research Computing at Hopkins core facility (rockfish.jhu.edu, supported by NSF grant number OAC1920103).

# Additional information

### Competing interests
Clifford P Brangwynne: C.P.B. is a founder and consultant for Nereid Therapeutics. The other authors declare that no competing interests exist.

### Funding

| Funder | Grant reference number | Author |
|---|---|---|
| National Science Foundation | PHY-1734030 | Yaojun Zhang<br>Andrew GT Pyo<br>Ned S Wingreen |
| National Institutes of Health | R01 GM140032 | Yaojun Zhang<br>Ned S Wingreen |
| Howard Hughes Medical Institute | | Clifford P Brangwynne |
| Air Force Office of Scientific Research | FA9550-20-1-0241 | Clifford P Brangwynne |
| Princeton Center for Complex Materials | DMR-1420541 | Clifford P Brangwynne<br>Howard A Stone |

The funders had no role in study design, data collection and interpretation, or the decision to submit the work for publication.

### Author contributions
Yaojun Zhang, Conceptualization, Resources, Data curation, Formal analysis, Supervision, Funding acquisition, Investigation, Methodology, Writing - original draft, Writing - review and editing; Andrew GT Pyo, Conceptualization, Methodology; Ross Kliegman, Data curation, Formal analysis, Writing - review and editing; Yoyo Jiang, Data curation, Formal analysis; Clifford P Brangwynne, Conceptualization, Supervision; Howard A Stone, Conceptualization, Supervision, Methodology, Writing - review and editing; Ned S Wingreen, Conceptualization, Resources, Formal analysis, Supervision, Funding acquisition, Investigation, Methodology, Writing - original draft, Writing - review and editing

### Author ORCIDs
Yaojun Zhang ⬤ https://orcid.org/0000-0003-4587-6834
Clifford P Brangwynne ⬤ https://orcid.org/0000-0002-1350-9960
Howard A Stone ⬤ https://orcid.org/0000-0002-9670-0639
Ned S Wingreen ⬤ https://orcid.org/0000-0001-7384-2821

Reviewer #1 (Public Review): https://doi.org/10.7554/eLife.91680.3.sa1
Author response https://doi.org/10.7554/eLife.91680.3.sa2

---

# Additional files

### Supplementary files
• MDAR checklist

• Source code 1. MATLAB scripts used to generate input files for the LAMMPS Molecular Dynamics Simulator, as well as the corresponding LAMMPS input files.

### Data availability
The current manuscript is a computational study. The codes for generating simulated data are uploaded as *Source code 1*.

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

## Appendix 1

### Derivation of the FRAP recovery curve in *Equations 5 and 6*

The exact solution for $f(t)$ in *Equation 4*, the fraction of molecules in a spherical condensate of radius $R$ which are unbleached at time $t$, is derived by *Taylor et al., 2019*, in an integral form using Laplace transforms (*Taylor et al., 2019*),

$$f(t) = 1 - \frac{6\alpha k^2}{\pi\sqrt{\lambda}} \int_0^\infty g(u, t^*)\, du, \tag{17}$$

where

$$g(u, t^*) = \frac{(u\cos u - \sin u)^2 \exp(-u^2 t^*)}{u^2 \left\{ u^2 \left[u\cos u + (k-1)\sin u\right]^2 + \lambda \left[(1 + k\frac{\alpha}{\lambda})u\cos u + (k - 1 - k\frac{\alpha}{\lambda})\sin u\right]^2 \right\}} \tag{18}$$

and $\lambda = D_{\text{dil}}/D_{\text{den}}$, $\alpha = c_{\text{den}}/c_{\text{dil}}$, $t^* = tD_{\text{den}}/R^2$, and $k = R\kappa/D_{\text{den}}$, where $\kappa$ is the interface conductance.

To obtain a more intuitive result, we first rearrange $g(u, t^*)$ as

$$g = \frac{\exp(-u^2 t^*)}{u^4 \left(1 + \frac{k}{u\cot u - 1}\right)^2 + \lambda u^2 \left(1 + k\frac{\alpha}{\lambda} + \frac{k}{u\cot u - 1}\right)^2}. \tag{19}$$

We note that the diffusion of biomolecules in the dilute phase is typically much faster than diffusion in the dense phase, i.e., $\lambda = D_{\text{dil}}/D_{\text{den}} \gg 1$. In this parameter regime, $g(u, t^*)$ is sharply peaked at the values of $u$ where

$$1 + k\frac{\alpha}{\lambda} + \frac{k}{u\cot u - 1} = 0, \tag{20}$$

i.e., when the second term in the denominator of $g(u, t^*)$ becomes 0. Representative $g(u, t^*)$ curves are shown in *Appendix 1—figure 1*. We can therefore approximate $g(u, t^*)$ as

$$g(u, t^*) = \sum_n a_n \frac{\lambda^2}{u_n^4 k^2 \alpha^2} \exp(-u_n^2 t^*)\delta(u - u_n), \tag{21}$$

where $u_n$ is the $n$ th solution of *Equation 20* and $a_n \sim 1/\sqrt{\lambda}$ is the inverse of effective width of $n$ th peak of $g(u, t^*)$. Clearly, the prefactor of the delta function $\delta(u - u_n)$ in *Equation 21* drops rapidly with increasing values of $u_n$. Consequently, the integral in *Equation 17* is always dominated by the contribution from the first mode, and therefore

$$f(t) \approx 1 - \exp(-t/\tau), \tag{22}$$

where

$$\tau = \frac{R^2}{D_{\text{den}} u_1^2}, \tag{23}$$

with $u_1$ the first root of *Equation 20*.

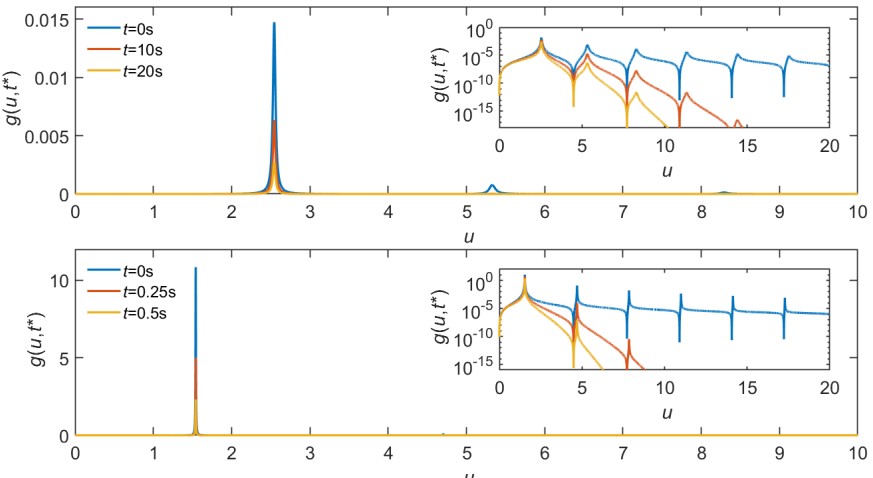

**Appendix 1—figure 1.** Representative curves of $g(u, t^*)$. Parameters matched to the simulated A6B6 system: $c_{den} = 7.7 \, mM$, $c_{dil} = 0.05 \, mM$, $D_{den} = 0.013 \, \mu m^2/s$, $D_{dil} = 17 \, \mu m^2/s$, and $\kappa = 0.14 \, \mu m/s$. Droplet radius is $R = 1 \, \mu m$ (top) and $0.1 \, \mu m$ (bottom). Inset: same plot with the $y$-axis in log scale.

At any given values of $k$, $\alpha$, and $\lambda$, the solutions of **Equation 20** can be obtained numerically. Alternatively, we can obtain an approximate analytical solution by first rewriting **Equation 20** as

$$1 - u \cot u = K = \frac{k}{1 + k\alpha/\lambda}. \tag{24}$$

We plot the combined parameter $K = k/(1 + k\alpha/\lambda)$ versus the first root $u_1$ in **Appendix 1—figure 2**. For small $K$, Taylor series expansion around $u = 0$ for the left side of **Equation 24** yields

$$1 - u \left( \frac{1}{u} - \frac{u}{3} \right) = \frac{u^2}{3} = K, \tag{25}$$

which yields $u_1 = \sqrt{3K}$. For large $K$, $u_1$ plateaus at $\pi$. We therefore approximate $u_1$ as

$$u_1 = \frac{\pi \sqrt{3K}}{\sqrt{\pi^2 + 3K}}. \tag{26}$$

This approximate solution is compared with the exact numerical solution of $u_1$ in **Appendix 1—figure 2**. The maximum error of about 5% occurs at an intermediate value of $K$ (**Appendix 1—figure 2**, inset). The relaxation time $\tau$ corresponding to the approximate solution in **Equation 26** is

$$\tau = \frac{R^2}{\pi^2 D_{den}} + \frac{c_{den} R^2}{3 c_{dil} D_{dil}} + \frac{R}{3\kappa}. \tag{27}$$

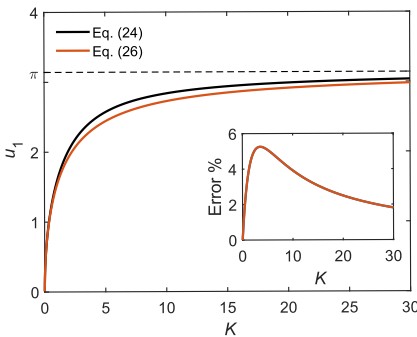

**Appendix 1—figure 2.** Comparison of exact and approximate solutions of **Equation 24**. Inset: percentage error of the approximate solution.

## Time required to replace all molecules in a spherical droplet in the absorbing boundary limit

The time evolution of the spherically symmetric concentration profile $c(r, t)$ of molecules in the dilute phase around a droplet of radius $R$ is given by

$$\frac{\partial}{\partial t} c(r, t) = D_{\text{dil}} \left( \frac{\partial^2}{\partial r^2} + \frac{2}{r} \frac{\partial}{\partial r} \right) c(r, t), \quad r > R. \tag{28}$$

If molecules incident from the dilute phase are immediately and irreversibly absorbed into the dense phase, the boundary condition is then $c(R, t) = 0$. The steady-state solution of **Equation 28** in this absorbing-boundary limit is

$$c(r) = c_{\text{dil}} \left( 1 - \frac{R}{r} \right), \quad r > R, \tag{29}$$

where $c_{\text{dil}}$ is the concentration at $r \to \infty$, which yields a total steady-state flux into the droplet of

$$J = -4\pi R^2 D_{\text{dil}} \left. \frac{d}{dr} c(r) \right|_{r=R} = 4\pi R D_{\text{dil}} c_{\text{dil}}. \tag{30}$$

It then takes a time

$$\tau = \frac{R^2 c_{\text{den}}}{3 D_{\text{dil}} c_{\text{dil}}} \tag{31}$$

to replace all $4\pi R^3 c_{\text{den}}/3$ molecules in the droplet.

## Derivation of the interface conductance in the continuum limit, yielding *Equations 8 and 13*

To derive the interface conductance in the continuum limit (which neglects the bouncing effect), we start with the mean-field formulation developed in *Hubatsch et al., 2021*, where the concentration of bleached components $c(\mathbf{r}, t)$ is governed by

$$\frac{\partial}{\partial t} c(\mathbf{r}, t) = -\nabla \cdot \mathbf{j}(\mathbf{r}, t), \tag{32}$$

with the flux

$$\mathbf{j}(\mathbf{r}, t) = -D \left[ c_{\text{eq}}(\mathbf{r}) \right] \left[ \nabla c(\mathbf{r}, t) - c(\mathbf{r}, t) \frac{\nabla c_{\text{eq}}(\mathbf{r})}{c_{\text{eq}}(\mathbf{r})} \right], \tag{33}$$

where $c_{eq}(\mathbf{r})$ is the equilibrium concentration profile, and $D[c_{eq}(\mathbf{r})]$ the diffusion coefficient which depends on the local equilibrium concentration.

For a spherical condensate of radius $R$, if the interface width is narrow, we can assume that the flux going through the interface is uniform in space along the radial direction, i.e.,

$$\boldsymbol{j}(\boldsymbol{r},t) = -D(r)\left[\frac{\partial}{\partial r}c(r,t) - \frac{c(r,t)}{c_{eq}(r)}\frac{d}{dr}c_{eq}(r)\right]\hat{r} = j(t)\hat{r},\tag{34}$$

where $\hat{r}$ denotes the unit vector in the radial direction. Therefore,

$$\frac{\partial}{\partial r}c(r,t) - \frac{c(r,t)}{c_{eq}(r)}\frac{d}{dr}c_{eq}(r) = -\frac{j(t)}{D(r)}.\tag{35}$$

The solution to the above equation is

$$c(r,t) = \frac{c_{eq}(r)c\left(R,t\right)}{c_{eq}\left(R\right)} - \int_R^r \frac{c_{eq}(r)j(t)}{c_{eq}(r')D(r')}dr'.\tag{36}$$

We assume that the interface spans a width of $\delta$ from $R_-$ to $R_+$ with $R_{+/-} = R \pm \delta/2$, then $c_{eq}\left(R - \delta/2\right) = c_{den}$ and $c_{eq}\left(R + \delta/2\right) = c_{dil}$. Substituting *Equation 36* into the second boundary condition in *Equation 3*, we have

$$j(t) = \kappa\left[c\left(R - \delta/2,t\right) - \frac{c_{den}}{c_{dil}}c\left(R + \delta/2,t\right)\right] = \kappa\int_{R-\delta/2}^{R+\delta/2}\frac{c_{den}j(t)}{c_{eq}(r)D(r)}dr.\tag{37}$$

We then obtain an expression for the interface conductance $\kappa$:

$$\kappa = \left[\int_{R-\delta/2}^{R+\delta/2}\frac{c_{den}}{c_{eq}(r)D(r)}dr\right]^{-1}.\tag{38}$$

The equilibrium concentration $c_{eq}(r)$ transitions from $c_{den}$ to $c_{dil}$ between $R - \delta/2$ and $R + \delta/2$, and the corresponding diffusion coefficient $D(r)$ transitions from $D_{den}$ to $D_{dil}$. Assuming a monotonic sigmoidal transition, along with $D_{den} \ll D_{dil}$ and $c_{den} \gg c_{dil}$, we obtain

$$\frac{c_{den}}{c_{eq}(r)D(r)} \leq \frac{1}{D_{den}} + \frac{c_{den}}{c_{dil}D_{dil}},\tag{39}$$

which leads to an interface conductance in the continuum limit

$$\kappa \geq \frac{1}{\delta}\left[\frac{1}{D_{den}} + \frac{c_{den}}{c_{dil}D_{dil}}\right]^{-1}.\tag{40}$$

In the simulations in *Figure 3*, we are only interested in molecules that remain in the dilute phase without entering the dense phase, and the corresponding interface conductance in the continuum limit is then

$$\kappa_0 = \frac{c_{dil}D_{dil}}{\delta c_{den}}.\tag{41}$$

## Derivation of the 1D, slab-geometry versions of *Equations 1–6*

For the case of a quasi-1D slab geometry, we consider the condensate to sit in the middle in the region $-l < x < l$ with the simulation box extending along the $x$-axis from $-L$ to $L$. The time evolution of the concentration profile $c(x,t)$ of molecules initially located in the dense phase (bleached population) is then given by the 1D diffusion equations:

$$\begin{aligned}\frac{\partial}{\partial t}c(x,t) &= D_{den}\frac{\partial^2}{\partial x^2}c(x,t), \quad |x| < l; \\ \frac{\partial}{\partial t}c(x,t) &= D_{dil}\frac{\partial^2}{\partial x^2}c(x,t), \quad l < |x| < L,\end{aligned}\tag{42}$$

with the initial condition:

$$c(x, 0) = \begin{cases} c_{\text{den}}, & |x| < l; \\ 0, & l < |x| < L, \end{cases} \tag{43}$$

and boundary conditions:

$$\frac{\partial}{\partial x} c(x, t)\Big|_{x=0} = \frac{\partial}{\partial x} c(x, t)\Big|_{x=L} = 0 \tag{44}$$

$$-D_{\text{den}} \frac{\partial}{\partial x} c(x, t)\Big|_{x=l_-} = -D_{\text{dil}} \frac{\partial}{\partial x} c(x, t)\Big|_{x=l_+} = \kappa \left[ c(l_-, t) - \frac{c_{\text{den}}}{c_{\text{dil}}} c(l_+, t) \right]. \tag{45}$$

The general solution for the diffusion *Equation 42* with the boundary condition in *Equation 44* is:

$$c(x, t) = \begin{cases} \sum_n a_n \cos(p_n x) e^{-D_{\text{den}} p_n^2 t}, & |x| < l; \\ \sum_n b_n \cos[q_n(L - x)] e^{-D_{\text{dil}} q_n^2 t}, & l < |x| < L. \end{cases} \tag{46}$$

Applying the boundary condition in *Equation 45* to this general solution yields:

$$D_{\text{den}} p_n^2 = D_{\text{dil}} q_n^2, \tag{47}$$

$$D_{\text{den}} a_n p_n \sin(p_n l) = -D_{\text{dil}} b_n q_n \sin(q_n(L - l)) = \kappa \left[ a_n \cos(p_n l) - \frac{c_{\text{den}}}{c_{\text{dil}}} b_n \cos\left( q_n(L - l) \right) \right], \tag{48}$$

which leads to

$$\frac{1}{\kappa} = \sqrt{\frac{\tau}{D_{\text{den}}}} \cot \frac{l}{\sqrt{\tau D_{\text{den}}}} + \frac{c_{\text{den}}}{c_{\text{dil}}} \sqrt{\frac{\tau}{D_{\text{dil}}}} \cot \frac{L - l}{\sqrt{\tau D_{\text{dil}}}}, \tag{49}$$

where $\tau = (D_{\text{den}} p_n^2)^{-1} = (D_{\text{dil}} q_n^2)^{-1}$ is the relaxation time of the $n$ th mode of the system. For given parameters $c_{\text{dil}}$, $c_{\text{den}}$, $D_{\text{dil}}$, $D_{\text{den}}$, $l$, $L$, and $\kappa$, $\tau$ can be obtained numerically using the above equation. In the regime where the interface conductance is small, we derive an analytical expression for the relaxation time

$$\tau = \frac{l(L - l)}{\kappa(L - l + c_{\text{den}} l / c_{\text{dil}})}, \tag{50}$$

which resembles the corresponding relaxation time for a spherical droplet when interface conductance is small $\tau = R/(3\kappa)$.

We note that, in principle, *Equation 49* can be used to infer the value of $\kappa$ for a system using the relaxation time $\tau$ from simulation. However, due to the relatively small simulation sizes, the interface regime can constitute a significant fraction of the dense-phase condensate. This can result in uncertainties in the determination of the dense-phase width $l$ and diffusion coefficient $D_{\text{den}}$, etc., leading to errors in the determination of the interface conductance using *Equation 49*. Such errors can be significant when slow diffusion in the dense phase becomes rate-limiting for overall system relaxation. Therefore, instead of sticking to the 'FRAP protocol', we find it more convenient to track the molecules that remain in the dilute phase without ever entering the dense phase (*Figure 3*), as this minimizes the errors caused by any inaccuracies in dense-phase parameters. To relate $\kappa$ to the decay time of the dilute-phase molecules, we note that the time evolution of the concentration profile $c(x, t)$ of the dilute-phase molecules which have never entered the dense phase is given by

$$\frac{\partial}{\partial t} c(x, t) = D_{\text{dil}} \frac{\partial^2}{\partial x^2} c(x, t), \qquad l < |x| < L, \tag{51}$$

with the initial condition $c(x, 0) = c_{\text{dil}}$ for $l < |x| < L$ and boundary conditions:

$$\frac{\partial}{\partial x} c(x,t)\bigg|_{x=L} = c(l_-,t) = 0;$$ (52)

$$-D_{\text{dil}} \frac{\partial}{\partial x} c(x,t)\bigg|_{x=l_+} = -\kappa \frac{c_{\text{den}}}{c_{\text{dil}}} c(l_+,t).$$ (53)

Going through a similar procedure as for *Equations 46–49*, we obtain

$$\kappa = \frac{c_{\text{dil}}}{c_{\text{den}}} \sqrt{\frac{D_{\text{dil}}}{\tau}} \tan \frac{L-l}{\sqrt{\tau D_{\text{dil}}}}.$$ (54)

The interface conductance $\kappa$ for simulated systems in *Figure 3C* (bottom) is obtained from the relationship in *Equation 54*, which is *Equation 12* in the main text, using the measured relaxation time $\tau$ (*Figure 3C*, top), of molecules that remain in the dilute phase without ever entering the dense phase.

## Derivation of the unbound-sticker parameter $u$, *Equations 14 and 15*

In the interface-resistance-dominated regime, the decay time $\tau$ of the number of dilute-phase molecules that have not entered the dense phase can be obtained from *Equation 54*:

$$\tau = \frac{c_{\text{dil}}(L-l)}{c_{\text{den}}\kappa}.$$ (55)

In the slab geometry, this decay time is controlled by the flux per unit area $j$ entering the dense phase

$$\tau = \frac{c_{\text{dil}}V}{2jA},$$ (56)

where $V$ is the volume of the dilute phase, and $A$ the cross-sectional area of the interface between the dilute and dense phases (the factor of 2 accounts for the two interfaces). Combining these two equations, we have

$$\kappa = \frac{j}{c_{\text{den}}}.$$ (57)

For our simulations of polymers with A and B type stickers, we can approximate $j$ by assuming that a polymer incident from the dilute phase will join the dense phase if and only if an unbound monomer on the polymer binds to an unbound monomer in the dense phase somewhere in the interface region. To find an approximate formula for $j$ we therefore need to estimate the rate of such binding events per unit area of the interface. To this end, we can use the formula for diffusion-limited monomer-monomer binding, but with some modifications: First, we can write the concentration of unbound monomers of type A in the dilute phase as

$$c_{\text{dilA}} = n_A c_{\text{dil}} f_{\text{dilA}},$$ (58)

where $n_A$ is the number of type A monomers per polymer and $f_{\text{dilA}}$ is the fraction of these monomers that are unbound. This concentration implies a diffusion-limited binding flux onto each unbound dense-phase type B monomer in the interface region

$$j_{A \to B} = 4\pi r_0 D_{\text{dil}} c_{\text{dilA}},$$ (59)

where $r_0$ is the sticker radius, and we have assumed that the diffusion rate is set by the whole polymer. Now we need an estimate for the areal density $\rho_{\text{denB, unbound}}$ of available unbound B-type monomers in the dense-phase interface region, since each one will contribute the above flux (*Equation 59*). We can write

$$\rho_{\text{denB, unbound}} = \delta n_B c_{\text{den}} f_{\text{denB}},$$ (60)

where $\delta$ is the width of the interface region, $n_B$ is the number of type B monomers per polymer, and $f_{\text{denB}}$ is the fraction of unbound B monomers on dense-phase polymers in this region. Finally, we can

combine the above equations, and include the binding of dilute-phase B-type monomers to dense-phase A-type monomers, to obtain

$$j = j_{A \to B} \rho_{denB, \, unbound} + j_{B \to A} \rho_{denA, \, unbound} = 4\pi r_0 D_{dil} \delta n_A n_B c_{dil} c_{den} \left( f_{dilA} f_{denB} + f_{dilB} f_{denA} \right). \quad (61)$$

The interface conductance $\kappa$ is then

$$\kappa = 4\pi r_0 D_{dil} \delta n_A n_B c_{dil} \left( f_{dilA} f_{denB} + f_{dilB} f_{denA} \right). \quad (62)$$

Using this expression, we can then estimate the ratio between the true $\kappa$ and its continuum limit $\kappa_0 = c_{dil} D_{dil} / (\delta c_{den})$ to be

$$\frac{\kappa}{\kappa_0} \approx \frac{4\pi r_0 \delta^2 n^2 c_{den}}{2 + s + s^{-1}} \left( f_{dilA} f_{denB} + f_{dilB} f_{denA} \right), \quad (63)$$

where $n = n_A + n_B$ is the length of a polymer and $s = c_A / c_B$ is the global stoichiometry. Note that we have used $n_A n_B = n^2 s / (1 + s)^2$ to derive the above expression. In practice, we find this expression to be quite accurate up to a constant prefactor (**Figure 3D**), and we define the right-hand side of **Equation 63** as a lumped parameter $u$.

## Appendix 2

### Simulation procedures and data recording

We perform coarse-grained molecular-dynamics simulations using LAMMPS (*Plimpton, 1995*) to simulate phase separation of 'sticker and spacer' polymers. Individual polymers are modeled as linear chains of spherical stickers of types A and B connected by implicit spacers (*Figure 2A*) with the interaction potentials in *Equations 9–11*, which ensure one-to-one binding between A and B stickers.

For each of the five selected sequences (*Figure 2A*), we simulate 1000 polymers in a 500 nm × 50 nm × 50 nm box with periodic boundary conditions. Following the simulation procedures in *Zhang et al., 2021*, we first initialize the simulation by confining polymers in the region $-90\,\text{nm} < x < 90\,\text{nm}$ to promote phase separation and ensure that only a single dense condensate is formed. The attractive interaction between A and B stickers (*Equation 10*) is gradually switched on from $U_0 = 0$ to 14 over $2.5 \times 10^7$ time steps. This annealing procedure leads to the formation of a dense phase close to its equilibrated concentration. The dense condensate is equilibrated at fixed $U_0 = 14$ for another $2.5 \times 10^7$ steps and then the confinement is removed. The system is equilibrated for $2 \times 10^8$ more time steps to allow for the formation of a dilute phase and further relaxation of the dense phase. We then record the positions of all particles every $2.5 \times 10^5$ steps for 800 recordings.

Through the entire simulation, we equilibrate the system using a Langevin thermostat implemented with LAMMPS commands *fix nve* and *fix langevin*, i.e., the system evolves according to *Langevin, 1908*

$$m\frac{d^2\vec{r}_i}{dt^2} = -\gamma\frac{d\vec{r}_i}{dt} - \nabla_{\vec{r}_i}U(\vec{r}_1, ..., \vec{r}_N) + \vec{f}, \tag{64}$$

where $\vec{r}_i$ is the coordinate of particle $i$, $m$ is its mass, $\gamma$ is the friction coefficient, $\vec{f}$ is random thermal noise, and the potential energy $U(\vec{r}_1, ..., \vec{r}_N)$ contains all interactions between particles, including bonds and sticker-sticker interactions (*Equations 9–11*). We take temperature $T = 300\,\text{K}$, damping factor $\tau = m/\gamma = 10\,\text{ns}$, step size $dt = 0.1\,\text{ns}$, and mass of particle $m = 188.5\,\text{ag}$. These parameters give each sticker the correct diffusion coefficient $D = k_\text{B}T/(3\pi\eta d)$, where $\eta$ is the water viscosity $0.001\,\text{kg/m/s}$ and $d = 2\,\text{nm}$ is the sticker diameter.

We perform 10 simulation replicates with different random seeds for each of the five selected polymer sequences. Consistency of results is checked across replicates. To test if the system has reached equilibrium, we compare the dense- and dilute-phase concentrations derived from the first and second halves of the recorded data. The agreement indicates that the system has reached equilibrium.

### Determining the dilute- and dense-phase concentrations

To measure the dilute- and dense-phase concentrations, we first group polymers into connected clusters in each recording. Two stickers are considered connected if they are part of the same polymer, or if they are within the attraction distance $r_0 = 1\,\text{nm}$. Connected stickers are then grouped into clusters. In all simulations, we observe one large cluster which contains most of the polymers, and tens to hundreds of very small clusters (*Figure 2B*). We consider the large cluster to constitute the dense phase, and the smaller clusters to be constituents of the dilute phase. To find the concentrations of each phase, we identify the center of mass of the dense cluster in each recording, and recenter the simulation box to this center of mass. We then compute the polymer concentration histogram along the $x$ axis with a bin size 1/50 of box length. The histogram of numbers of stickers per bin is averaged over all recordings and simulation replicates. The polymer concentration profile is derived as the sticker concentration profile divided by the number of stickers per polymer. The resulting polymer concentration profile has high values in the middle corresponding to the dense-phase concentration, and low values on the two sides corresponding to the dilute-phase concentration (*Figure 2C*). The dilute- and dense-phase concentrations in *Figure 2D* are calculated by averaging the concentration profile over the regions ($x \le -150\,\text{nm}$ or $x \ge 150\,\text{nm}$) and ($-10\,\text{nm} \le x \le 10\,\text{nm}$), respectively.

### Determining the dilute- and dense-phase diffusion coefficients

To measure the dilute- and dense-phase diffusion coefficients, we perform simulations with a pure dilute phase or dense phase, i.e., with polymers at the measured dilute- and dense-phase concentrations. Specifically, for the dilute-phase case, we simulate 750 $(A2B2)_3$, 285 $(A3B3)_2$, 101

A6B6, 104 A8B6, and 180 A10B6 polymers, each in a $150\,\text{nm} \times 150\,\text{nm} \times 150\,\text{nm}$ box with periodic boundary conditions. For the dense-phase case, we simulate 1000 polymers for all selected sequences in a $W\,\text{nm} \times 50\,\text{nm} \times 50\,\text{nm}$ box with periodic boundary conditions, where $W = 116.1$ for $(A2B2)_3$, 96.5 for $(A3B3)_2$, 85.8 for A6B6, 136.5 for A8B6, and 234.2 for A10B6. To equilibrate the system, the attractive interaction between A and B stickers (*Equation 10*) is gradually switched on from $U_0 = 0$ to $14\,k_\text{B}T$ over $2.5 \times 10^7$ time steps and equilibrated at fixed $U_0 = 14\,k_\text{B}T$ for $2 \times 10^8$ more time steps. We then record the displacement of all particles every $2.5 \times 10^5$ steps for 400 recordings. Five simulation replicates with different random seeds are performed for each selected sequence.

To find the diffusion coefficients, we compute the time-averaged MSD for each polymer as a function of the lag time $t_\text{lag}$, and average over all polymers in a simulation box and over five replicates. The time- and ensemble-averaged MSD is then linearly fit to $\text{MSD} = 6Dt_\text{lag}$ to extract the diffusion coefficient.

## Determining the interface conductance

To measure the interface conductance $\kappa$, we follow the simple protocol depicted in *Figure 3A*. This scheme, based on the rate that particles in the dilute phase join the dense phase, is both computationally efficient and allows us to infer the interface conductance even when slow diffusion in the dense phase is rate-limiting for overall system relaxation. Specifically, in this protocol we first define a 'survival' variable $S$ for each polymer as a function of time: $S = 1$ if the polymer belongs to any dilute-phase cluster (including a solo cluster), and $S = 0$ if the polymer is in the dense-phase cluster. Next, for all polymers starting with $S = 1$ (i.e. in the dilute phase), we check if there is a period of time (chosen here to be 10 times the average bond lifetime of an isolated A-B pair) for which its $S$ value is always 0 (i.e. the polymer has joined the dense-phase cluster). If yes, we set $S = 1$ at the time points before the joining event and $S = 0$ at all times afterward. If not, we set $S = 1$ for this polymer for all time points. We then average $S(t)$ over all polymers starting with $S = 1$ and over the 10 simulation replicates. The obtained $S(t)$ is the average survival probability of polymers in the dilute phase that have never entered the dense phase. We fit $S(t)$ to a decaying exponential to extract the decay time $\tau$. The interface conductance $\kappa$ is then calculated using *Equation 54* with the measured decay time and dilute- and dense-phase parameters.

In *Figure 3*, we set the criterion for a polymer to have entered the dense phase as being continuously connected to the dense-phase cluster for a duration longer than $10\tau$, where $\tau$ is the average bond lifetime of an isolated A-B sticker pair. Briefly, the bond lifetime of an isolated pair is obtained by simulating a bound pair of A-B stickers in a box and recording the time when they first separate by the cutoff distance of the attractive interaction $r_0 = 1\,\text{nm}$. The mean bond lifetime $\tau$ is found by averaging results of 1000 replicates with different random seeds. In *Appendix 1—figure 1*, we compare the results for the interface conductance $\kappa$ using alternative durations, $> \tau$ and $> 20\tau$, as criteria for joining the dense phase. The value of $\kappa$ changes very little between the $10\tau$ and $20\tau$ criteria, suggesting that the results in *Figure 3* are robust to the definition of 'joining' the dense phase, provided very short-lived bonds are neglected.

As an alternative approach, we calculated $\kappa$ by directly measuring the flux $j$ of molecules that enter the dense phase and then using $\kappa = j/c_\text{den}$ (*Equation 57*). To find this flux, we first define an entering event as occurring when a molecule starting from the dilute phase joins the dense-phase cluster and stays for a duration longer than 10 times the average bond lifetime for an isolated A-B sticker pair. We count the number of total entering events $N$ in a simulation (note that some molecule can enter the dense phase multiple times), and the flux is then $N/(2AT)$, where $A$ is the cross-sectional area of the interface and $T$ is the duration of the simulation. We show in *Appendix 1—figure 2* that the values of $\kappa$ obtained via this method are consistent with the results reported in *Figure 3C*.

We show in *Appendix 2—figure 3*, a few representative trajectories of A6B6 (top) and A10B6 (bottom) polymers 'bouncing' off the interface between dilute and dense phases. More bouncing events per unit time are observed in the A6B6 system compared to A10B6 system, consistent with the presence of a larger interface resistance in the A6B6 system.

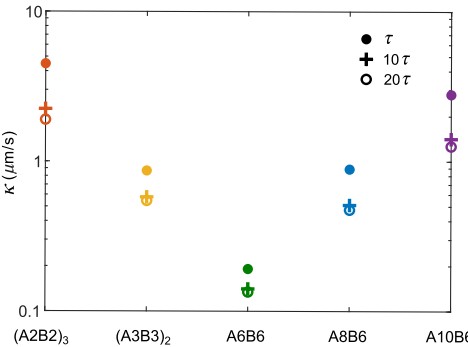

**Appendix 2—figure 1.** Interface conductance inferred from average survival time of particles initially in the dilute phase for different criteria for having 'joined' the dense phase (*Figure 3*). The interface conductance is shown for three criteria: a polymer is considered to have joined the dense phase if it is in the dense-phase cluster for a continuous duration of $\tau$, $10\tau$, or $20\tau$, where $\tau$ is the average bond lifetime of an isolated A-B sticker pair.

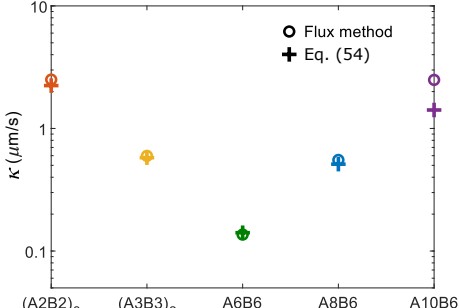

**Appendix 2—figure 2.** Comparison of interface conductance $\kappa$ obtained with the flux method and with *Equation 54* as reported in *Figure 3C*.

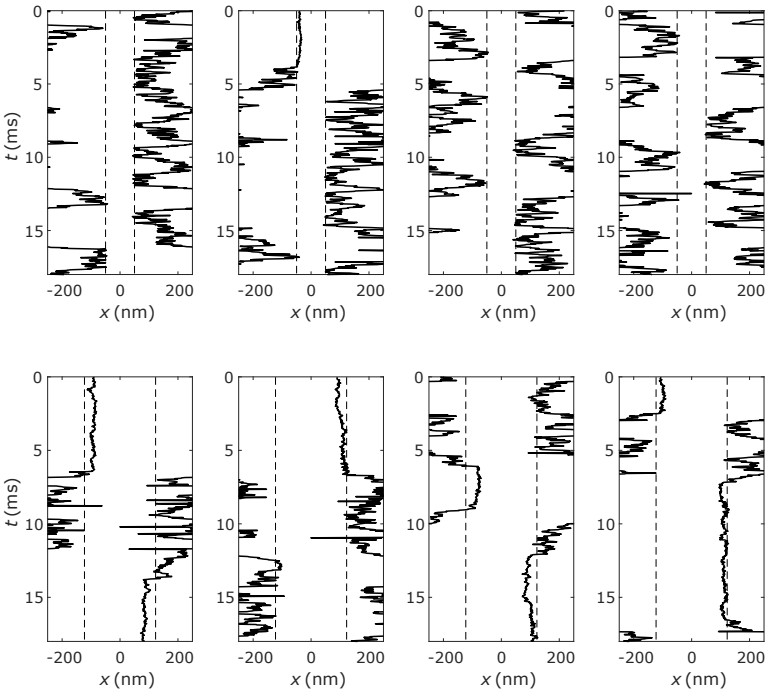

**Appendix 2—figure 3.** Representative trajectories of molecules 'bouncing' multiple times at the interface of the A6B6 system (top) and A10B6 system (bottom).

## Details of simulation and theory of FRAP recovery of an A6B6 droplet

Simulations of the A6B6 droplet system largely follow their counterparts in the slab geometry. 2000 polymers were placed inside a cubic box with periodic boundary conditions. Half of the polymers are initially confined in a sphere of radius 55 nm and the other half kept outside. The attraction between A and B stickers is gradually turned on from $U_0 = 0$ to 14 over $4 \times 10^7$ time steps. The system is equilibrated at fixed $U_0 = 14$ for another $1 \times 10^7$ steps and then the spherical confinement is removed. The above procedures ensure the formation of a single droplet and a uniform dilute phase at a desired concentration. We started with simulation boxes of side lengths 300 nm and 275 nm, which correspond to initial dilute-phase concentrations of 0.06 mM and 0.08 mM. The droplets shrank and grew accordingly over time in these simulations, which allowed us to extrapolate to the correct box size of side length 286 nm for a stable droplet (*Appendix 2—figure 4*). The system is equilibrated for $5 \times 10^7$ more time steps to allow equilibration between the dilute and dense phases. We then record the positions of all particles every $1 \times 10^6$ steps for 600 recordings. 10 simulation replicates were performed.

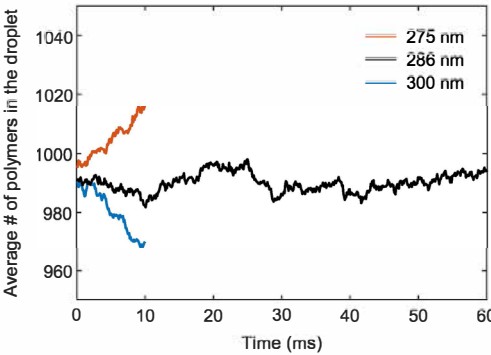

**Appendix 2—figure 4.** A simulation box of side length 286 nm stabilizes the droplet size. Number of polymers in the droplet as a function of time averaged over 10 simulation replicates for simulation boxes of side length 275 nm, 286 nm, and 300 nm.

To obtain the concentration profile in *Figure 4B*, we identify the center of mass of the droplet and recenter the simulation box to this center of mass in each recording. We then compute the time- and ensemble-averaged polymer concentration histogram along the radial direction with a bin size 5.5 nm. The dilute- and dense-phase concentrations ($c_{\mathrm{dil}}^{\mathrm{d}}$ and $c_{\mathrm{den}}^{\mathrm{d}}$) of the droplet system are calculated by averaging the concentration profile over the regions $r \geq 55\,\mathrm{nm}$ and $r \leq 12.5\,\mathrm{nm}$, respectively. To obtain the concentration profile of the bleached population at time $t$ after photobleaching in *Figure 4C*, we label all polymers in the droplet at time $t_0$ as bleached and track the concentration profile of these polymers at a later time $t_0 + t$. Results are averaged over all possible choices of $t_0$ and over 10 simulation replicates. Interface conductance of the droplet system is determined using the flux method. We count the total number of entering events, $N$, in which a polymer starts from the dilute phase and joins the droplet for a duration longer than $10\tau$ ($6 \times 10^6$ steps). $\kappa^{\mathrm{d}}$ is then $N/(AT)$, where $A$ is the surface area of the droplet and $T$ is the duration of the simulation.

To obtain the theory curve in *Figure 4D*, we numerically integrate *Equation 1* using a finite-difference method. We used the modified initial and boundary conditions:

$$c(r, 0) = \begin{cases} c_{\mathrm{den}}^{\mathrm{d}}, & r \leq R; \\ 0, & R < r \leq R_{\mathrm{box}}, \end{cases} \tag{65}$$

and

$$\left.\frac{\partial}{\partial r} c(r, t)\right|_{r=0} = \left.\frac{\partial}{\partial r} c(r, t)\right|_{r=R_{\mathrm{box}}} = 0, \tag{66}$$

to account for the system's finite size with parameters: $c_{\mathrm{den}}^{\mathrm{d}} = 8.1\,\mathrm{mM}$, $c_{\mathrm{dil}}^{\mathrm{d}} = 0.073\,\mathrm{mM}$, $D_{\mathrm{den}} = 0.013\,\mu\mathrm{m}^2/\mathrm{s}$, $D_{\mathrm{dil}} = 17\,\mu\mathrm{m}^2/\mathrm{s}$, $R = 0.037\,\mu\mathrm{m}$, and $\kappa^{\mathrm{d}} = 0.20\,\mu\mathrm{m}/\mathrm{s}$ directly extracted from simulations, and a spherical confinement of radius $R_{\mathrm{box}} = 0.177\,\mu\mathrm{m}$, which corresponds to the same volume as the cubic simulation box. *Equation 1* was integrated over time with a forward Euler scheme with a radial step of $0.5\,\mathrm{nm}$ and a time step of $5\,\mathrm{ns}$ to ensure numerical stability and accuracy. The resulting bleached concentration profile is then used in conjunction with *Equation 4* to obtain the FRAP recovery curve.

To derive *Equation 16* in the main text, we note that in the interface-resistance-dominant regime diffusion in the dilute and dense phases are both fast, therefore the bleached molecules can be assumed to have uniform concentration profiles, $c_{\mathrm{in}}(t)$ and $c_{\mathrm{out}}(t)$, inside and outside of the droplet, respectively. The net outward flux across the interface can then be written as $j(t) = \kappa[c_{\mathrm{in}}(t) - (c_{\mathrm{den}}/c_{\mathrm{dil}})c_{\mathrm{out}}(t)]$, which reduces the total number of bleached molecules inside the droplet according to:

$$\frac{4\pi R^3}{3} \frac{dc_{\mathrm{in}}(t)}{dt} = -4\pi R^2 j(t). \tag{67}$$

The above equation together with the total number of bleached molecules

$$\frac{4\pi R^3}{3} c_{\text{den}} = \frac{4\pi R^3}{3} c_{\text{in}}(t) + \left( \frac{4\pi R_{\text{box}}^3}{3} - \frac{4\pi R^3}{3} \right) c_{\text{out}}(t) \tag{68}$$

and the initial condition $c_{\text{in}}(0) = c_{\text{den}}$ can be solved to yield an analytical solution for $c_{\text{in}}(t)$:

$$c_{\text{in}}(t) = c_{\text{den}} \left[ 1 - A \left( 1 - e^{-\frac{t}{A\tau}} \right) \right], \tag{69}$$

where

$$A = \frac{\left( R_{\text{box}}^3 - R^3 \right) c_{\text{dil}}}{\left( R_{\text{box}}^3 - R^3 \right) c_{\text{dil}} + R^3 c_{\text{den}}} \tag{70}$$

corresponds to the maximum recovery intensity and $\tau = R/(3\kappa)$ corresponds to the recovery time for a system of infinite size in the interface-resistance-dominant regime. The FRAP recovery curve is then

$$f(t) = 1 - \frac{c_{\text{in}}(t)}{c_{\text{den}}} = A \left( 1 - e^{-\frac{t}{A\tau}} \right). \tag{71}$$

We note that in the limit where the system is infinite in size ($R_{\text{box}} \to \infty$), we recover the familiar fluorescence recovery curve in **Equation 5**.

