## [Editor Report · eLife assessment]

This **valuable** contribution studies factors that impact molecular exchange between dense and dilute phases of biomolecular condensates through continuum models and coarse-grained simulations. The authors provide **convincing** evidence that the bouncing of molecules off the interface can lead to interfacial resistance and limit mixing. Results like these can inform how experimental results in the field of biological condensates are interpreted.

---

## [Referee Report · Reviewer #1 (Public Review)]

Summary:

In this paper by Zhang, the authors build a physical framework to probe the mechanisms that underlie exchange of molecules between coexisting dense and dilute liquid-like phases of condensates. They first propose a continuum model, in the context of a FRAP-like experiment where the fluorescently labeled molecules inside the condensate are bleached at t=0 and the recovery of fluorescence is measured. Through this model, they identify how the key timescales of internal molecular mixing, replenishment from dilute phase, and interface transfer contribute to molecular exchange timescale. Motivated by a recent experiment reported by some of the co-authors previously (Brangwynne et al. in 2019) finding strong interfacial resistance in in vitro protein droplets of LAF-1, they seek to understand the microscopic features contributing to the interfacial conductance (inversely proportional to the resistance). To check, they perform coarse-grained MD-simulations of sticker-spacer self-associative polymers and report how conductance varies significantly even across the few explored sequences. Further, by looking at individual trajectories, they postulate the "bouncing" i.e., molecules that approach the interface but are not successfully absorbed is a strong contributor to this mass transfer limitation. Consistent with their predictions, sequences that have more free unbound stickers (i.e., for example through imbalance sequence sticker stoichiometries) have higher conductances and they show a simple linear scaling between number of unbound stickers and conductance. Finally, they predict that an droplet-size dependent transition in recovery time behavior.

Strengths:

(1) This paper is overall well-written and clear to understand.

(2) By combining coarse-grained simulations, continuum modeling, and comparison to published data, the authors provide a solid picture of how their proposed framework relates to molecular exchange mechanisms that are dominated by interface resistance and LAF-1 droplets.

(3) The choice of different ways to estimate conductance from simulation and reported data are thoughtful and convincing on their near-agreement (although a little discussion of why and when they differ would be merited as well).

Updated re-review:

This revised update by Zhang et al. is improved and addresses many of the concerns raised by myself and the other reviewer, especially with the expanded discussion, contextualized text in model description, and the addition of a nice example case-study in revised Fig. 4. I believe the paper provides solid evidence of how "bouncing" may contribute to interfacial resistance/exchange dynamics in biomolecular condensates and is a useful study for the community.

Note: In their response, the authors bring up an important point in references for LAF1 mutant FRAP data. While I found a few papers, for example https://www.pnas.org/doi/abs/10.1073/pnas.2000223117 and https://www.cell.com/biophysj/fulltext/S0006-3495(23)00464-2, these are likely to be not whole droplet bleaches. I wonder whether it may be possible to approximately predict the conductance from other parameters (such as from effective expressions in eq 14) to roughly estimate what the effect maybe since LAF-1 has fairly "known" stickers and spacers. Note that this is not required at all, but I just bring this up in case it may be of interest to authors!

---

## [Author Response]

The following is the authors’ response to the original reviews.

**eLife assessment**
This valuable contribution studies factors that impact molecular exchange between dense and dilute phases of biomolecular condensates through continuum models and coarse-grained simulations. The authors provide solid evidence that interfacial resistance can cause molecules to bounce off the interface and limit mixing. Results like these can inform how experimental results in the field of biological condensates are interpreted.

We would like to sincerely thank the editors for spending time on our manuscript and for the very positive assessment of our work. We have carefully considered and addressed the reviewers’ comments in the point-by-point response below and have revised our manuscript accordingly.

**Reviewer #1 (Public Review):**
Summary:In this paper by Zhang, the authors build a physical framework to probe the mechanisms that underlie the exchange of molecules between coexisting dense and dilute liquid-like phases of condensates. They first propose a continuum model, in the context of a FRAP-like experiment where the fluorescently labeled molecules inside the condensate are bleached at t=0 and the recovery of fluorescence is measured. Through this model, they identify how the key timescales of internal molecular mixing, replenishment from dilute phase, and interface transfer contribute to molecular exchange timescale. Motivated by a recent experiment reported by some of the co-authors previously (Brangwynne et al. in 2019) finding strong interfacial resistance in in-vitro protein droplets of LAF-1, they seek to understand the microscopic features contributing to the interfacial conductance (inversely proportional to the resistance). To check, they perform coarse-grained MD simulations of sticker-spacer self-associative polymers and report how conductance varies significantly even across the few explored sequences. Further, by looking at individual trajectories, they postulate that "bouncing" - i.e., molecules that approach the interface but are not successfully absorbed - is a strong contributor to this mass transfer limitation. Consistent with their predictions, sequences that have more free unbound stickers (i.e., for example through imbalance sequence sticker stoichiometries) have higher conductances and they show a simple linear scaling between the number of unbound stickers and conductance. Finally, they predict a droplet-size-dependent transition in recovery time behavior.Strengths:(1) This paper is well-written overall and clear to understand.(2) By combining coarse-grained simulations, continuum modeling, and comparison to published data, the authors provide a solid picture of how their proposed framework relates to molecular exchange mechanisms that are dominated by interface resistance and LAF-1 droplets.(3) The choice of different ways to estimate conductance from simulation and reported data are thoughtful and convincing in their near agreement (although a little discussion of why and when they differ would be merited as well).

We would like to thank the reviewer for the positive evaluation of our work. Indeed, we are grateful to the reviewer for this thoughtful, detailed, and constructive report, which has helped us strengthen the manuscript.

Weaknesses:(1) Almost the entirety of this paper is motivated by a previously reported FRAP experiment on a particular LAF-1 droplet in vitro. There are a few major concerns I have with how the original data is used, how these results may generalize, and the lack of connection of predictions with any other experiments (published or new).a. The mean values of cdense, cdilute, diffusivities, etc. are taken from Taylor et al. to rule in the importance of interfacial mass transfer limits. While this may be true, the values originally inferred (in the 2019 paper that this paper is strongly built off) report extremely large confidence intervals/inferred standard errors. The authors should accordingly report all their inferences with correct standardized errors or confidence intervals, which in turn, allow us to better understand these data.

Yes, agreed. We have now included the standard errors of the parameters from Taylor et al. (2019), and reported the corresponding standard errors for the timescales and interface conductance using error propagation. We have modified Fig. 1C right panel as well as the text in the figure caption:

“(Right) Expected recovery times τdil=cdenR2/(3cdilDdil)=4.2±3.2 s and τden =R2/(π2Dden)=60±18 s if the slowest recovery process was either the flux from the dilute phase or diffusion within the droplet, respectively, with Dden =0.0017±0.0005μm2/s,Ddil =94±11μm2/s and Taylor et al. (2019). While the timescale associated with interface resistance cden/cdil=1190±880 taken from τint  is unknown, the measured recovery time τ=4567±471 s is much longer than τdil and τden , suggesting the recovery is limited by flux through the interface, with an interface conductance of κ=R/3/(τ−τdil −τden)=(7.4±0.8)×10−5μm/s (Below Figure 1)”

b. The generalizability of this model is hard to gauge when all comparisons are made to a single experiment reported in a previous paper.i. Conceptually, the model is limited to single-component sticker-spacer polymers undergoing phase separation which is already a very simplified model of condensates - for e.g., LAF1 droplets in the cell have no perceptible interfacial mass limitations, also reported in Taylor et al. 2019 - so how these mechanisms relate to living systems as opposed to specific biochemistry experiments. So the authors need to discuss the implications and limitations of their model in the living context where there are multiple species, finite-size effects, and active processes at play.

We thank the reviewer for the critical comment. To address this point, we have included a paragraph in the Discussion regarding in vivo situations:

“In this work, we focused on the exchange dynamics of in vitro single-component condensates. How is the picture modified for condensates inside cells? It has been shown that Ddx4-YFP droplets in the cell nucleus exhibit negligible interface resistance *Taylor et al.* (*2019*), which raises the question whether interface resistance is relevant to natural condensates in vivo. Future quantitative FRAP and single-molecule tracking experiments on different types of droplets in the cell will address this question. One complication is that condensates in cells are almost always multi-component, which can increase the complexity of the exchange dynamics. Interestingly, formation of multiple layers or the presence of excess molecules of one species coating the droplet is likely to increase interface resistance. A notable example is the Pickering effect, in which adsorbed particles partially cover the interface, thereby reducing the accessible area and the overall condensate surface tension, slowing down the exchange dynamics *Folkmann et al.* (*2021*). The development of theory and modeling for the exchange dynamics of multi-component condensates is currently underway. (Lines 323-334)”

ii. Second, can the authors connect their model to make predictions of the impact of perturbations to LAF-1 on exchange timescales? For example, are mutants (which change the number or positioning of "stickers") expected to show particular trends in conductances or FRAP timescales? Since LAF-1 is a relatively well-studied protein in vitro, can the authors further contrast their expectations with already published datasets that explore these perturbations, even if they don't generate new data?

Our model is intended to address interface exchange dynamics at the conceptual level. The underlying mechanism for the large interface resistance of LAF-1 droplets could be more complicated than explored in our work. To study the impact of perturbations to LAF-1 on exchange timescales likely requires substantially more sophisticated molecular dynamics simulations. We undertook an extensive search for FRAP experiments on LAF-1 droplets where the whole droplet is photobleached, but were not able to find another dataset. We would be grateful if the reviewer is aware of such data and can point us to it.

iii. A key prediction of the interface limitation model is the size-dependent crossover in FRAP dynamics. Can the authors reanalyze published data on LAF-1 (albeit of different-size droplets) to check their predictions? At the least, is the crossover radius within experimentally testable limits?

Based on our prediction, the crossover radius for LAF-1 droplet is around 70 𝜇m. We have added a sentence in the text to point this out:

“We also predict the crossover for LAF-1 droplets to be around 𝑅 = 71 𝜇m, which in principle can be tested experimentally. (Lines 285-286)”

Unfortunately, most of FRAP experiments in Taylor at al. (2019) are partial FRAP experiments, in which only part of the dense phase is photobleached. The recovery time for such experiments reflects primarily the internal mixing speed of the dense phase rather than the exchange dynamics at the interface or transport from the dilute phase.

c. The authors nicely relate the exchange timescale to various model parameters. Is LAF-1 the only protein for which the various dilute/dense concentrations/diffusivities are known? Given the large number of FRAP and other related studies, can the authors report on a few other model condensate protein systems? This will help broaden the reach of this model in the context of other previously reported data. If such data are lacking, a discussion of this would be important.

Yes, indeed, we have found numerous publications with FRAP experiments performed on whole droplets of various proteins. However, none of these have provided a complete set of parameters to allow a quantitative analysis. Part of the reason is because it is nontrivial to have an accurate measurement of the partition coefficient (cden/cdil). We have added a sentence in the Discussion to promote future quantitative experiment and analysis of condensate exchange dynamics:

“We hope that our study will motivate further experimental investigations into the anomalous exchange dynamics of LAF-1 droplets and potentially other condensates, and the mechanisms underlying interface resistance. (Lines 320-322)”

To broaden the audience for this work in the hope of stimulating such studies, we have also modified the title and abstract so that it will be more visible to the FRAP community:

“The exchange dynamics of biomolecular condensates (Line 1)”

“A hallmark of biomolecular condensates formed via liquid-liquid phase separation is that they dynamically exchange material with their surroundings, and this process can be crucial to condensate function. Intuitively, the rate of exchange can be limited by the flux from the dilute phase or by the mixing speed in the dense phase. Surprisingly, a recent experiment suggests that exchange can also be limited by the dynamics at the droplet interface, implying the existence of an “interface resistance”. Here, we first derive an analytical expression for the timescale of condensate material exchange, which clearly conveys the physical factors controlling exchange dynamics. We then utilize sticker-spacer polymer models to show that interface resistance can arise when incident molecules transiently touch the interface without entering the dense phase, i.e., the molecules “bounce” from the interface. Our work provides insight into condensate exchange dynamics, with implications for both natural and synthetic systems. (Lines 16-26)”

(2) The reported sticker-spacer simulations, while interesting, represent a very small portion of the parameter space. Can the authors - through a combination of simulation, analyses, or physical reasoning, comment on how the features of their underlying microscopic model (sequence length, implicit linker length, relative stoichiometry of A/B for a given length, overall concentration, sequence pattern properties like correlation length) connect to conductance? This will provide more compelling evidence relating their studies beyond the cursory examination of handpicked sequences. A more verbose description of some of the methods would be appreciated as well, including specifically how to (a) calculate the bond lifetime of isolated A-B pair, and (b) how equilibration/convergence of MD simulations is established.

In our simulation, the interface conductance is essentially controlled by the fraction of unbound stickers, the encounter rate of a pair of unbound stickers, the dilute- and dense-phase concentrations, and the width of the interface. As a result, weaker binding strength and/or deviation of A:B stoichiometry from 1:1 result in a higher interface conductance. A6B6 polymers with long blocks of stickers of the same type (compared to (A2B2)3 and (A3B3)2) have a lower dilute-phase concentration and thinner interface width, so lower conductance. Sequence length and implicit linker length can have more complex effects, which are beyond the scope of the current study. We have now provided an explicit expression for 𝜅 in Equation (14) and added a discussion sentence in the text:

“More generally, we find that the interface conductance of the sticker-spacer polymers is controlled by the encounter rate of a pair of unbound stickers and the availability of these stickers, which in turn depends on the sticker-sticker binding strength, the dilute- and dense-phase polymer concentrations, and the width of the interface:κ=4πr0δDdiln2cdil2+s+s−1(fdilfdenB+fdilBfdenA)

where *n* is the number of monomers in a polymer, is the global stoichiometry (i.e., cA/cB), fdilA / dilB  and fden A/den⁡B are the fractions of unbound A/B monomers in the dilute and dense phases. (Lines 208-214)”

We have also added a few sentences in Appendix 2 to describe how we calculate the bond lifetime of an isolated A-B pair and how equilibration in simulations is established.

“Briefly, the bond lifetime of an isolated pair is obtained by simulating a bound pair of A-B stickers in a box and recording the time when they first separate by the cutoff distance of the attractive interaction nm. The mean bond lifetime 𝜏 is found by averaging results of 1000 replicates with different random seeds. (Lines 642-645)”

“To test if the system has reached equilibrium, we compare the dense- and dilute-phase concentrations derived from the first and second halves of the recorded data. The agreement indicates that the system has reached equilibrium. (Lines 586-589)”

(3) A lot of the main text repeats previously published models (continuum ones in Taylor et al. 2019 and Hubsatch et al., 2021, amongst others) and the idea of interface resistance being limiting was already explored quantitatively in Taylor 2019 (including approximate estimates of mass transfer limitations) - this is fine in context. While the authors do a good job of referring to past work in context, the main results of this paper, in my reading, are:- a simplified physical form relating conductance timescales.- sticker-spacer simulations probing microscopic origins.- analysis of size-dependent FRAP scaling.I am stating this not as a major weakness, but, rather - I would recommend summarizing and categorizing the sections to make the distinctions between previously reported work and current advances sufficiently clear.

We thank the reviewer for a clear summary of the contributions of our work. We have highlighted our main contributions in multiple places:

“Here, we first derive an analytical expression for the timescale of condensate material exchange, which clearly conveys the physical factors controlling exchange dynamics. We then utilize sticker-spacer polymer models to show that interface resistance can arise when incident molecules transiently touch the interface without entering the dense phase, i.e., the molecules “bounce” from the interface. (Lines 21-25)”

“In the following, we first derive an analytical expression for the timescale of condensate material exchange, which conveys a clear physical picture of what controls this timescale. We then utilize a “sticker-spacer” polymer model to investigate the mechanism of interface resistance. We find that a large interface resistance can occur when molecules bounce off the interface rather than being directly absorbed. We finally discuss characteristic features of the FRAP recovery pattern of droplets when the exchange dynamics is limited by different factors. (Lines 65-70)”

“Specifically, we first derived an analytical expression for the exchange rate, which conveys the clear physical picture that this rate can be limited by the flux of molecules from the dilute phase, by the speed of mixing inside the dense phase, or by the dynamics of molecules at the droplet interface. Motivated by recent FRAP measurements *Taylor et al.* (*2019*) that the exchange rate of LAF-1 droplets can be limited by interface resistance, which contradicts predictions of conventional mean-field theory, we investigated possible physical mechanisms underlying interface resistance using a “sticker-spacer” model. Specifically, we demonstrated via simulations a notable example in which incident molecules have formed all possible internal bonds, and thus bounce from the interface, giving rise to a large interface resistance. Finally, we discussed the signatures in FRAP recovery patterns of the presence of a large interface resistance. (Lines 291-300)”

**Reviewer #2 (Public Review):**
Summary:In this paper, the authors have obtained an analytical expression that provides intuition about regimes of interfacial resistance that depend on droplet size. Additionally, through simulations, the authors provide microscopic insight into the arrangement of sticky and non-sticky functional groups at the interface. The authors introduce bouncing dynamics for rationalizing quantity recovery timescales.I found several sections that felt incomplete or needed revision and additional data to support the central claim and make the paper self-contained and coherent.

We thank the reviewer for spending time on our manuscript and for the helpful critical comments.

First, the analytical theory operates with diffusion coefficients for dilute and dense phases. For the dilute phase, this is fine. For the dense phase, I have doubts that dynamics can be described as diffusive. Most likely, dynamics is highly subdiffusive due to crowded, entangled, and viscoelastic environments of densely packed interactive biomolecules. Some explanation and justification are in order here.

The reviewer is correct in noting that molecules within a condensate can move subdiffusively due to the viscoelastic nature of the condensate. However, subdiffusion only occurs at short time and small length scales, the motion of molecules becomes diffusive at longer time and larger length scales. The crossover time here is the terminal relaxation time measured to be on the order of milliseconds to seconds for typical condensates (see *Alshareedah, Ibraheem, et al.* "Determinants of viscoelasticity and flow activation energy in biomolecular condensates" Science Advances 10.7, 2024). We previously have also found that, for sticker-spacer polymers, this relaxation time is determined by the time it takes for a sticker to switch to a new partner (see *Ronceray et al.* (2022) in References), which is therefore largely determined by the bond lifetime of a sticker pair. The crossover length scale is expected to be comparable to the size of a molecule based on the theory of polymer disentanglement. Importantly, in order for the bleached droplet to recover its fluorescence, the bleached molecules must travel for a much longer time and a much larger length than the crossover time and length. It is therefore expected that the molecules move diffusively on the relevant timescale of a FRAP experiment, albeit with a diffusion coefficient that reflects crowding and entanglement on short time and length scales.

The second major issue is that I did not find a clean comparison of simulations with the derived analytical expression. Simulations test various microscopic properties on the value of k, which is important. But how do we know that it is the same quantity that appears in the expressions? Also, how can we be sure that analytical expressions can guide simulations and experiments as claimed? The authors should provide sound evidence of the predictive aspect of their derived expressions.

We thank the reviewer for raising this critical issue. We agree with the reviewer that we did not perform an explicit simulation to validate the developed theory, which leaves a gap between our theory and simulations. The main reason is because simulation of an *in silico* “FRAP experiment” on a 3D droplet is very computationally costly. Nevertheless, following the reviewer’s suggestion, we have now performed such a simulation in which we “bleached” a small A6B6 droplet and measured its recovery time. The good agreement between simulation and theory helps validate our overall combined computational and analytical approach. We have incorporated the new simulation and results into the manuscript. Two new sections including new figures (Figure 4 and Appendix 2 Figure 4) are added: “Direct simulation of droplet FRAP” in the main text (lines 232-261) and “Details of simulation and theory of FRAP recovery of an A6B6 droplet” in Appendix 2 (lines 665-715).

Are the plots in Figure 4 coming from experiment, theory, and simulation? I could not find any information either in the text or in the caption.

Figure 4 (now Figure 5) is from theory which uses parameters of the A6B6 system in simulation. We have added the following sentences to clarify:

“We compare the measured FRAP recovery time for the small droplet (green circle) to theoretical predictions from Equation (6) (gray) and Equations (1) - (4) (black) in Figure 5A. (Lines 255-257)”

“Figure 5. FRAP recovery patterns for large versus small droplets can be notably different for condensates with a sufficiently large interface resistance. (A) Expected relaxation time as a function of droplet radius for *in silico* “FRAP experiments” on the A6B6 system. The interface resistance dominates recovery times for smaller droplets, whereas dense-phase diffusion dominates recovery times for larger droplets. Green circle: FRAP recovery time obtained from direct simulation of an A6B6 droplet of radius 37 nm. Black curve: the recovery time as a function of droplet radius from a single exponential fit of the exact solution of the recovery curve from Equations (1) - (4). Gray curve: the recovery time predicted by Equation (6). Yellow, blue, and red curves: the recovery time when dense-phase, dilute-phase, and interface flux limit the exchange dynamics, i.e., the first, second, and last term in Equation (6), respectively. Parameters matched to the simulated A6B6 system in the slab geometry: (B) Time courses of fluorescence profiles for A6B6 droplets of radius (top) and (bottom); red is fully bleached, green is fully recovered. These concentration profiles are the numerical solutions of Equations (1) - (3) using the parameters in (A). (Below Figure 5)”